# Offshore wind energy forecasting sensitivity to sea surface temperature input in the Mid-Atlantic

Stephanie Redfern[1], Mike Optis[1], Geng Xia[1], and Caroline Draxl[1]

[1]National Renewable Energy Laboratory, Golden, Colorado, USA

**Correspondence:** Stephanie Redfern (stephanie.redfern@nrel.gov)

**Abstract.** As offshore wind farm development expands, accurate wind resource forecasting over the ocean is needed. One important yet relatively unexplored aspect of offshore wind resource assessment is the role of sea surface temperature (SST). Models are generally forced with reanalysis data sets, which employ daily SST products. Compared with observations, significant variations in SSTs that occur on finer time scales are often not captured. Consequently, shorter-lived events such as sea breezes and low-level jets (among others), which are influenced by SSTs, may not be correctly represented in model results. The use of hourly SST products may improve the forecasting of these events. In this study, we examine the sensitivity of model output from the Weather Research and Forecasting Model (WRF) 4.2.1 to different SST products. We first evaluate three different data sets: the Multiscale Ultrahigh Resolution (MUR25) SST analysis, a daily, 0.25° x 0.25° resolution product; the Operational Sea Surface Temperature and Ice Analysis (OSTIA), a daily, 0.054° x 0.054° resolution product; and SSTs from the Geostationary Operational Environmental Satellite 16 (GOES-16), an hourly, 0.02° x 0.02° resolution product. GOES-16 is not processed at the same level as OSTIA and MUR25; therefore, the product requires gap-filling using an interpolation method to create a complete map with no missing data points. OSTIA and GOES-16 SSTs validate markedly better against buoy observations than MUR25, so these two products are selected for use with model simulations while MUR25 is at this point removed from consideration. We run the model for June and July of 2020 and find that for this time period, in the Mid-Atlantic, although OSTIA SSTs overall validate better against in situ observations taken via a buoy array in the area, the two products result in comparable hub-height (140-m) wind characterization performance on monthly time scales. Additionally, during hours-long flagged events (< 30 h each) that show statistically significant wind speed deviations between the two simulations, both simulations once again demonstrate similar validation performance (differences in bias, earth mover's distance, correlation, and root mean square error on the order of $10^{-1}$ or less), with GOES-16 winds validating nominally better than OSTIA winds. With a more refined GOES-16 product, which has been not only gap-filled but also assimilated with in situ SST measurements in the region, it is likely that hub-height winds characterized by GOES-16-informed simulations would definitively validate better than those informed by OSTIA SSTs.

*Copyright statement.* This work was authored in part by the National Renewable Energy Laboratory, operated by Alliance for Sustainable Energy, LLC, for the U.S. Department of Energy (DOE) under Contract No. DE-AC36-08GO28308. Funding provided by the U.S. Department of Energy Office of Energy Efficiency and Renewable Energy Wind Energy Technologies Office and by the National Offshore Wind

Research and Development Consortium under Agreement No. CRD-19-16351. The views expressed in the article do not necessarily represent the views of the DOE or the U.S. Government. The U.S. Government retains and the publisher, by accepting the article for publication, acknowledges that the U.S. Government retains a nonexclusive, paid-up, irrevocable, worldwide license to publish or reproduce the published form of this work, or allow others to do so, for U.S. Government purposes.

## 1  Introduction

The United States Atlantic coast is a development site for upcoming offshore wind projects. There are 15 leasing areas located throughout the Atlantic Outer Continental Shelf, where a number of offshore wind farms are planned to be developed (Bureau of Ocean Energy Management, 2018). Therefore, characterizing offshore boundary layer winds in the region has risen in importance. Accurate forecasting will provide developers with a better understanding of local wind patterns, which can inform wind farm planning and layout decisions (Banta et al., 2018). Additionally, improved weather prediction will allow for real-time adjustments of turbine operation to increase their operating efficiency and protect them against unnecessary wear and tear (Gutierrez et al., 2016, 2017; Debnath et al., 2021).

The Mid-Atlantic Bight (MAB) is an offshore cold pool region spanning the eastern United States coast from North Carolina up through Cape Cod, Massachusetts, and it overlies the offshore wind leasing areas. The cold pool forms during the summer months, when the ocean becomes strongly stratified and the thermocline traps colder water near the ocean floor. During the transition to winter, as sea surface temperatures (SSTs) drop, the stratification weakens and the cold pool breaks down. Thus, the cold pool generally persists from the spring through the fall. Southerly winds that drive surface currents offshore will result in coastal upwelling of this colder water. And, at times, strong winds associated with storm development can mix the cold pool upward, cooling the surface and influencing near-surface temperatures and winds (Colle and Novak, 2010; Chen et al., 2018; Murphy et al., 2021).

Accurate representation of the MAB in forecasting models is important because SSTs are closely tied to offshore winds. Horizontal temperature gradients between land and the ocean, as well as vertical temperature gradients over the ocean—which can form, for example, when SSTs are anomalously cold, as with the MAB—help define offshore airflow. In particular, variations in temperature can lead to or impact short-lived offshore events occurring on hourly time scales, such as sea breezes and low-level jets (LLJs). Sea breezes are driven by the land-sea temperature difference, which, if strong enough (around 5°C or greater), can generate a circulation between the water and the land (Stull et al., 2015). With a relatively colder ocean, as during summer months, this leads to a near-ground breeze blowing landward, with a weak recirculation toward the ocean aloft (Miller et al., 2003; Lombardo et al., 2018). Similarly, the near-surface horizontal and air-sea temperature differences dictate the strength of stratification over the ocean. Studies have found a robust link between atmospheric stability and LLJ development, so accurately representing SSTs is key to modeling near-surface stability and, accordingly, LLJs (Gerber et al., 1989; Kaellstrand, 1998; Kikuchi et al., 2020; Debnath et al., 2021). A WRF analysis of the region conducted by Aird et al. (2022) found that in the MAB leasing areas specifically, LLJs, occurred on 12% of the hours in June, 2010-2011. Both LLJs and sea

breezes can affect individual wind turbine and whole farm operation, so forecasting them correctly can improve power output and turbine reliability (Nunalee and Basu, 2014; Pichugina et al., 2017; Murphy et al., 2020; Xia et al., 2021).

Typical climate and weather model initialization and forcing inputs are reanalysis products, such as ERA5 and MERRA-2, which are global data sets that assimilate model output with observations to create a comprehensive picture of climate at each time step considered (Gelaro et al., 2017; Hersbach et al., 2020). These data sets primarily include global SST products that are produced at lower temporal and spatial resolutions than what can be available via regional, geostationary satellites. These coarser-resolution data sets, therefore, do not capture observed hourly and, in many cases, diurnal fluctuations in SSTs,

which may influence their ability to properly force sea breezes and LLJs. Some preliminary comparisons between weather simulations, forced with different SST products, indicate that this particular input can have a significant impact on modeled offshore wind speeds (Byun et al., 2007; Chen et al., 2011; Dragaud et al., 2019; Kikuchi et al., 2020).

Few studies have examined the impact of finer-temporal resolution SST products specifically on wind forecasting, and to the authors' knowledge, none so far have focused on the Mid-Atlantic. There have been studies looking at numerical weather

prediction model (NWP) sensitivity to SST, but they have considered other regions or different, often coarser spatial and temporal resolution, products (Chen et al., 2011; Park et al., 2011; Shimada et al., 2015; Dragaud et al., 2019; Kikuchi et al., 2020; Li et al., 2021). In this article, we explore the effects of forcing the Weather Research and Forecasting Model (WRF), a NWP used for research and operational weather forecasting, with different SST data sets characterized by different spatial and temporal resolutions, in the Mid-Atlantic region during the summer months. Specifically, we address differences in model

performance on monthly time scales and then contrast characterization effectiveness during shorter wind events. Section 2 lays out the data, model setup, and methods used in this study. Section 3 explains the findings of our simulations, and Section 4 explores their implications. Finally, Section 5 summarizes the intent of the study as well as its findings.

## 2    Methods

We first validate three different SST data sets against observations taken at an array of buoys off the Mid-Atlantic coast during

the months of June and July, 2020. Following validation, we select two of the three data sets for use as inputs to two different model simulations in the MAB region, which are identically configured aside from the SST data. August is not considered due to data availability constraints at the time of the study. The output data are compared with in situ measurements taken at buoys (SSTs) and floating lidars (winds) in the region. A 140-m hub height is assumed, based on the analysis of regions with moderate wind resource by Lantz et al. (2019). We evaluate performance primarily via a set of validation metrics calculated on

monthly time scales. We then flag specific events during which the model generally captures regional winds, but output from the two simulations deviate significantly (defined in this study as one or more standard deviations from their mean differences) from one another. Again, validation analysis is performed for these periods.

**Table 1.** Buoy and lidar locations, owner, and data availability.

| Buoy or Lidar | Name | Latitude (°) | Longitude (°) | SST Depth (m) | % Data Available | Owner |
|---|---|---|---|---|---|---|
| Lidar & Buoy | E05 | 40.1614 | -72.7396 | 0.8 | 100.0 | NYSERDA |
| Lidar & Buoy | E06 | 39.6273 | -73.4123 | 0.8 | 100.0 | NYSERDA |
| Lidar & Buoy | ASOW-6 | 39.2717 | -73.8892 | 1 | 84.89 | Atlantic Shores |
| Buoy | 44017 | 40.693 | -72.049 | 1.5 | 96.32 | NDBC |
| Buoy | 44025 | 40.251 | -73.164 | 1.5 | 96.46 | NDBC |
| Buoy | 44065 | 40.369 | -73.703 | 1.5 | 96.51 | NDBC |
| Buoy | 44075 | 40.363 | -70.883 | 1 | 29.66 | Ocean Observatories Initiative |
| Buoy | 44076 | 40.137 | -70.775 | 1 | 29.58 | Ocean Observatories Initiative |
| Buoy | 44077 | 39.940 | -70.883 | 1 | 100.0 | Ocean Observatories Initiative |
| Buoy | 44091 | 39.768 | -73.770 | 0.46 | 100.0 | U.S. Army Corps of Engineers |
| Buoy | 44097 | 40.967 | -71.126 | 0.46 | 100.0 | Ocean Observatories Initiative |

## 2.1 In Situ and Lidar Data

This study makes use of both SST and wind profile observational data for model validation. SSTs are provided by the National Buoy Data Center (NBDC) at several locations along the Mid-Atlantic Coast, as listed in Table 1 and shown in Fig. 1. Buoy data located at the Atlantic Shores Offshore Wind ASOW-6 location are also used (Fig. 1b). Wind data have been taken from the Atlantic Shores Offshore Wind floating lidar and the two New York State Energy Research & Development Authority (NYSERDA) floating lidars, whose locations are listed in Table 1. These lidars provide wind speed and wind direction at 10-minute intervals from either 10 m (Atlantic Shores) or 40 m (NYSERDA) up through 250 m above sea level. There are periods of missing data for all buoys and lidars.

## 2.2 Model Setup

WRF Version 4.2.1 is the NWP employed in this study (Powers et al., 2017). WRF is a fully compressible, non-hydrostatic model that is used for both research and operational applications. Our model setup, including key physics and dynamics options, are outlined in Table 2.

The study area spans the majority of the MAB, with the nested domain (grid spacing of 2 km x 2 km) running from the mid-Virginia coast to the south up through Cape Cod to the north (Fig. 1a).

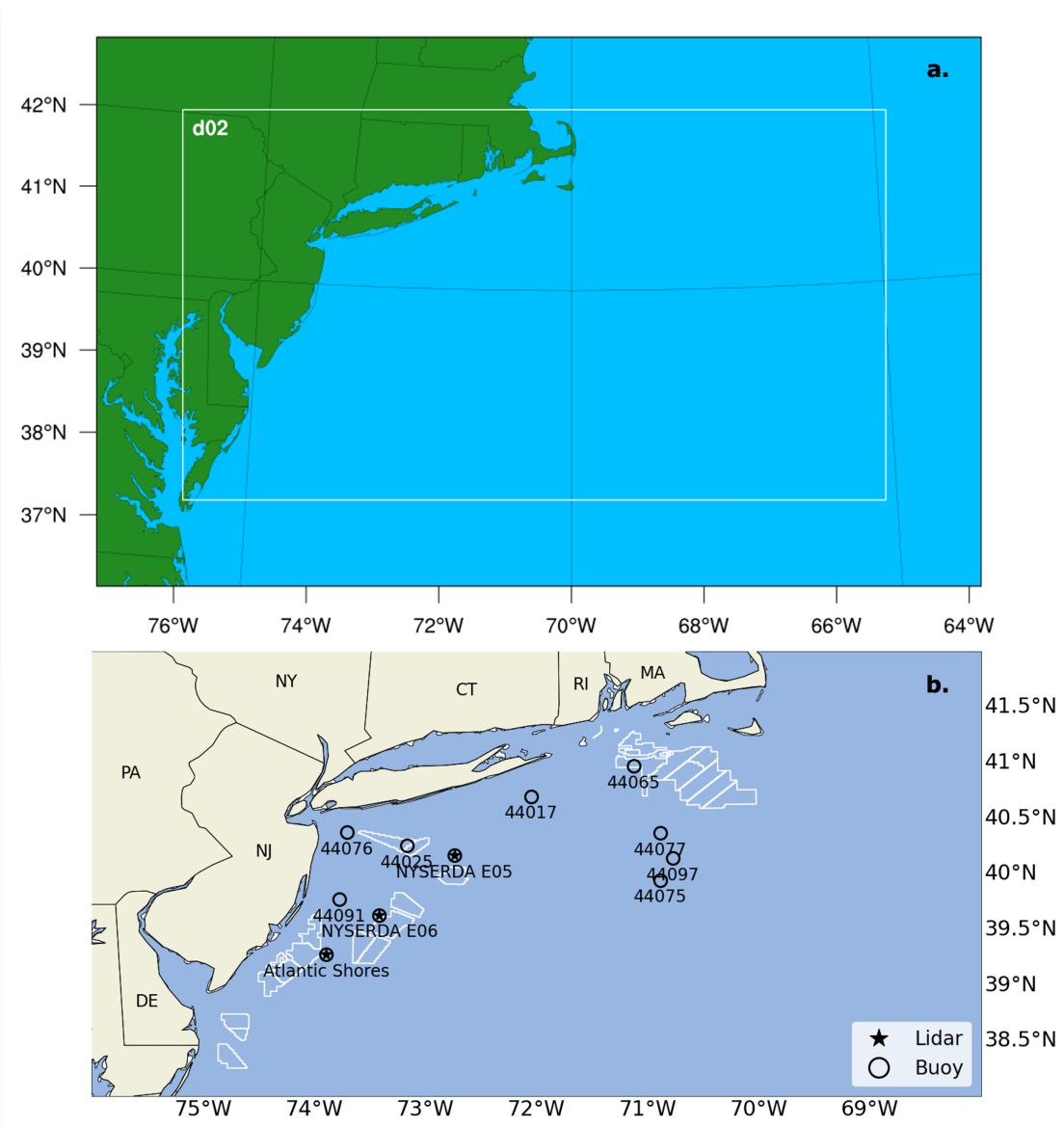

**Figure 1.** The WRF domains (a.) and a zoomed in, more detailed look at a subset of the nested domain, with markers indicating the lidars (stars) and buoys (hollow circles) that were used in this study (b.). The Atlantic Shores and two NYSERDA locations each host both a buoy and a lidar. Planning areas are outlined in white.

**Table 2.** Model setup.

| WRF Parameter | Selection |
|---|---|
| Number of Domains | 2 |
| Domain Resolution | 6 km (Parent), 2 km (Nest) |
| Output Time Resolution | 5 minutes |
| Vertical Levels | 61 |
| Reanalysis Data | ERA5 (Hersbach et al., 2020) |
| Microphysics | Ferrier (Schoenberg Ferrier, 1994) |
| Radiation Scheme | RRTMG (longwave & shortwave) (Iacono et al., 2008) |
| Planetary Boundary Layer | Nakanishi and Niino (MYNN) (Nakanishi and Niino, 2006) |
| Surface Layer Parameterization | MYNN (Nakanishi and Niino, 2009) |
| Land Surface Scheme | Unified Noah land-surface model (Tewari et al., 2004) |
| Cumulus Parameterization | Kain-Fritsch (Kain and Fritsch, 1993) |
| Upper-Level Damping | Rayleigh at 5km depth |

## 2.3 Sea Surface Temperature Data

We compare how well three different SST datasets validate against buoy observations and subsequently select the two best-performing data sets to force our simulations (Table 3). Aside from these different SST product inputs, the rest of the model parameters in the simulations remain identical.

We have selected the ERA5 global reanalysis data set to force our simulations. OSTIA is the SST data set native to this product (Hersbach et al., 2020). As such, when included as part of ERA5, OSTIA's resolution has been adjusted to match ERA5's 31 km spatial resolution and hourly temporal resolution. For our simulations, however, we overwrite these SSTs with OSTIA data at its original resolution of $0.05°$, MUR25 data at its $0.25°$ resolution, and GOES-16 data at its resolution of $0.02°$.

The coarsest-resolution product we consider is the $0.25°$ Multi-scale Ultra-high Resolution SST analysis (MUR25). MUR25 is a daily product that assimilates observations and model data to render a complete global grid with SSTs. MUR25 has undergone pre- and post-processing and the SSTs in the dataset are foundation temperatures, which are measured deep enough in the water to discount diurnal temperature fluctuations. MUR25 has a spatial resolution of $0.25°$ x $0.25°$ (Chin et al., 2017), which is significantly coarser than either of the other two SST data sets we evaluate.

Our next SST selection, the Operational Sea Surface Temperature and Sea Ice Analysis (OSTIA) system, is a daily global product that combines in situ observations taken from buoys and ships, model output, and multiple remotely sensed SST data sets (Stark et al., 2007; Donlon et al., 2012). Similar to MUR25, it is a complete product in that it has no missing data. OSTIA has a spatial resolution of $0.05°$ x $0.05°$. Additionally, as a daily product like MUR25, OSTIA provides foundation SSTs (Stark et al., 2007; Donlon et al., 2012; Fiedler et al., 2019).

**Table 3.** SST data sets used in this study.

| Parameter | MUR25 | OSTIA | GOES-16 |
|---|---|---|---|
| Satellite Coverage | Global | Global | Regional |
| Temporal Resolution | Daily | Daily | Hourly |
| Spatial Resolution | 0.25° | 0.054° | 0.02° |
| Temperature Type | Foundation | Foundation | SST |
| Processing Level | Gridded & assimilated with in situ observations | Gridded & assimilated with in situ observations | Gridded only |
| Gap-Filling | Released product is filled | Released product is filled | DINEOF needed |

Our finest-resolution product is taken via GOES-16, which is a geostationary, regional satellite with a spatial resolution of 0.02° x 0.02° (Schmit et al., 2005, 2017). This product does not assimilate its measurements with in situ data. While GOES-16 does not offer global coverage and, therefore, cannot be used for certain world regions, it does cover the Mid-Atlantic Bureau of Ocean Energy Management (BOEM) offshore wind lease areas, which is our region of interest. Because GOES-16 has an hourly resolution, it can capture diurnal changes in SST and, therefore, measures surface temperature (Schmit et al., 2005, 2008).

Due to the lesser level of processing in the GOES-16 data set, it contains numerous data gaps that must be filled. This missing data is the result of a post-processing algorithm, which flags pixels that fall below a specified temperature threshold, that is applied prior to release of the data set. This filter is in place to remove cloud cover. While this method is effective with regard to its defined intent, it can also erroneously discard valid pixels that capture the cold water upwelling typical to the MAB region during the warmer months. And, although this cold-pixel filter is also a common practice in global SST data sets (OSTIA), the high level of post-processing applied in those products interpolates over and fills the missing grid cells prior to release.

To gap-fill the GOES-16 data so that we may use it in our application, we employ the Data INterpolating Emperical Orthogonal Function algorithm, or DINEOF, which is an open-source application that applies empirical orthogonal function (EOF) analysis to reconstruct incomplete data sets (Ping et al., 2016). The program was originally designed to specifically gap-fill remotely observed SSTs that contain missing data due to cloud-flagging and removal algorithms (as is the case with the GOES-16 data), and has demonstrated strong results in past studies (Alvera-Azcárate et al., 2005; Ping et al., 2016).

We additionally include in the GOES-16 SST data set the sensor-specific error statistics (SSES) bias field that is included with the distributed product. This component accounts for retrieval bias using a statistical algorithm designed to correct for errors in the SST field. Compared with the SST values alone, the bias-corrected GOES-16 data offsets an inherent warm bias in the raw data.

**Table 4.** Error metrics considered in this study.

| Error Metric | Equation |
|---|---|
| Bias | $\bar{p} - \bar{o}$ |
| Unbiased Root Mean Square Error (RMSE) | $\left[ \frac{1}{N} \sum_{n=1}^{N} [(p_n - \bar{p})(o_n - \bar{o})]^2 \right]^{\frac{1}{2}}$ |
| Square of Correlation Coefficient ($R^2$) | $\left[ \frac{\frac{1}{N} \sum_{n=1}^{N} (p_n - \bar{p})(o_n - \bar{o})}{\sigma_p \sigma_o} \right]^2$ |
| Wasserstein Metric / Earth-Mover's Distance (EMD) | $\sum_{i=1}^{m} \sum_{j=1}^{n} M_{ij} d_{ij}$ |

## 2.4 Event Selection

We are particularly interested in understanding how well the model forecasts shorter wind events, as LLJs and sea breezes occur on hourly time scales. We have created a set of parameters that, when met, detect relatively brief time periods (on the order of hours to days) during which one simulation may be outperforming the other, which we then more closely examine to evaluate differences in their wind profile characterization and SST validation.

For an event to be flagged, it must meet the following criteria:

1. Correlation for both models is above 0.5 at hub height, for two of the three lidar locations.

2. Differences in wind speeds between the two models must be greater than one standard deviation from the monthly mean difference.

3. Gaps during which the wind speed difference drops below one standard deviation must not persist for more than 2 hours during a single event.

4. Events must last for at least one hour.

This set of event characteristics first acts to filter out periods during which WRF is generally under-performing—possibly due to model shortcomings outside of SST forcing—so that the performance difference in the selected events may be with more certainty attributed to SSTs. Then, events are located during which the two simulations forecast statistically significantly different hub-height wind speeds, which persist for a period of time long enough to substantially affect power generation.

## 2.5 Validation Metrics

To evaluate which simulation performs best during our study period, we calculate sets of validation metrics, as outlined by Optis et al. (2020). Specifically, we look at SSTs and 140-m (hub-height) wind speeds, on both monthly and event time scales. The metrics we calculate are named and defined in Table 4, and they give a quantification of the error present in each case.

The bias provides information on the average simulation performance during the evaluation period—specifically, if the model is consistently over- or under-predicting the output variable in consideration. Unbiased root mean square error (RMSE)

provides a more nuanced look at the spread of the error in the results. The square of the correlation coefficient (referred to from here forward as correlation) quantifies how well the simulations' variables change in coordination with those of the observations. And, finally, the Wasserstein Metric, also known as the Earth-Mover's Distance (EMD), measures the difference between the observed and simulated variable distributions.

## 3   Results

We first assess how well each SST data set compares with buoy data. Following SST validation, we evaluate the model's wind characterization performance when forced with different SST data. We assume a hub height of 140 m and compare output winds at this altitude against measurements taken via floating lidars off the Mid-Atlantic coast. Specifically, we analyze monthly performance and then select several shorter periods during June and July 2020 during which we compare wind characterization accuracy.

After evaluation of the SST data sets, only OSTIA- and GOES-16-forced WRF simulations are compared, as during July—one half of our study period—MUR25 SSTs validate significantly worse against buoy measurements than the other two products. OSTIA SSTs validate the best out of the three data sets. Average, simulated hub-height winds across monthly periods validate similarly for both OSTIA and GOES-16. At an event-scale temporal resolution—that is, on the order of hours—GOES-16 and OSTIA perform comparably, with GOES-16 marginally outperforming OSTIA.

### 3.1   Sea Surface Temperature Performance

We evaluate SST performance by linearly interpolating the satellite-based products to 10-minute intervals (the in situ data output resolution) and making a brief qualitative assessment followed by calculating and comparing at each buoy validation metrics for the different products.

The time series plots in Fig. 2 show that that while GOES-16 tracks the diurnal cycle seen by observations, the other two products do not (the highlighted time frames in the figure are wind events that are discussed in Section 3.3). Despite this feature, however, there exist a number of periods each month when GOES-16 does not accurately capture observed dips in temperature. And, during many of these times, the daily data sets, though missing the nuance of GOES-16, better represent the colder SSTs.

A comparison of all three products show markedly poorer validation against the floating lidar buoys, for at least two metrics at each buoy, by MUR25 as compared with GOES-16 and OSTIA during July 2020 (Table 5). Because July 2020 is one half of our study period, we remove MUR25 from further analysis.

Over the course of June and July combined, looking across the entire buoy array, OSTIA overall outperforms GOES-16, as shown in Fig. 3. Both products have a relatively strong cold bias (between -0.1°C and -0.25°C) compared with observations at the three lidar locations. GOES-16 presents a negative bias at five additional buoys, and OSTIA has a negative bias at one other buoy. Both SST products display warm biases at buoys 44017 (off the northeast coast of Long Island) and 44076 (the farthest offshore location considered, southeast of Cape Cod). Although GOES-16 follows the diurnal cycle rather than representing

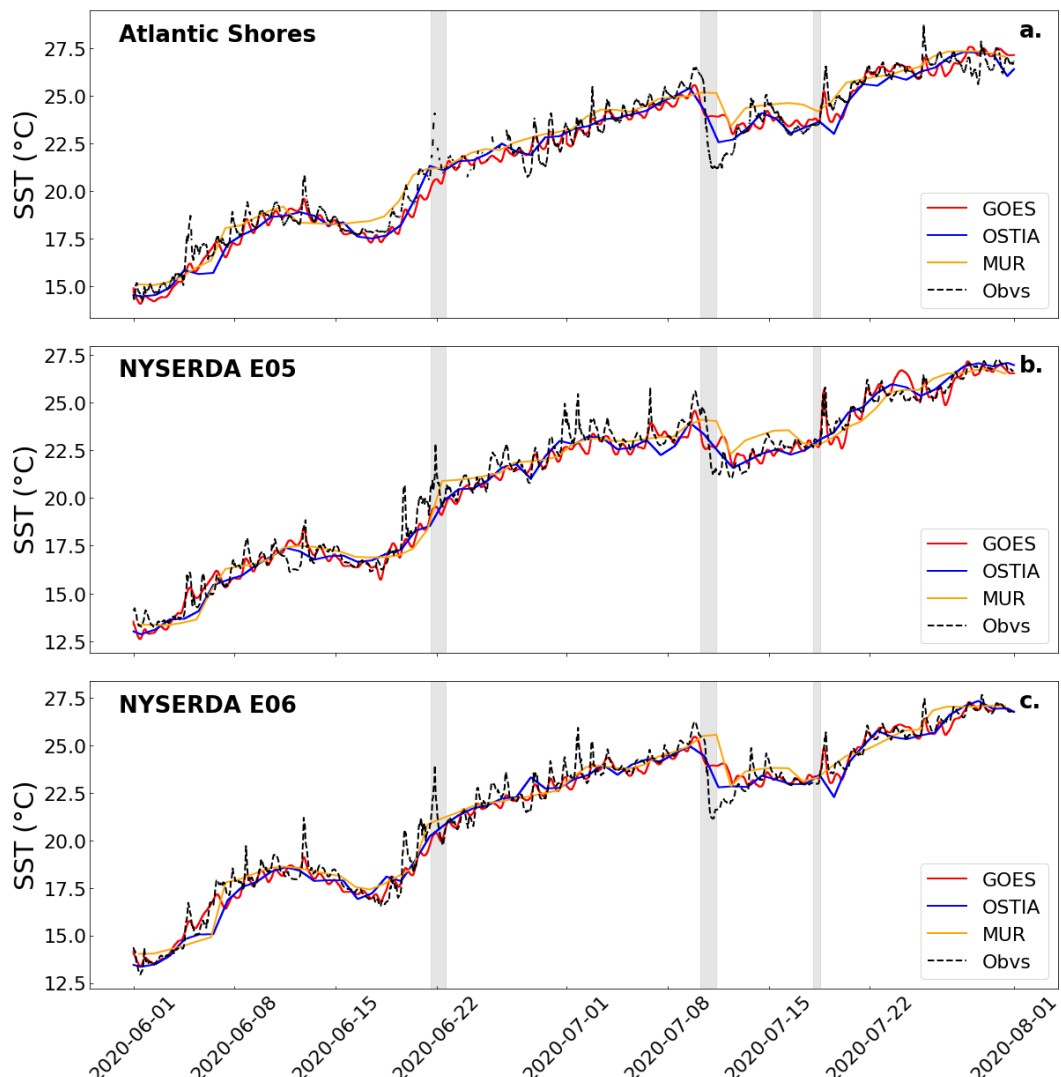

**Figure 2.** Time series of MUR25, OSTIA, GOES-16, and in situ measurements of SST at the Atlantic Shores buoy for June and July at Atlantic Shores (a.), NYSERDA E05 (b.), and NYSERDA E06 (c.). Specific wind events that are evaluated in Section 3.3 are highlighted in gray.

only the daily average SSTs, as is the case with OSTIA, on average the correlation across all sites is comparable between the two SST products. RMSE is higher for GOES-16 at every buoy (with an average $0.64°C$, compared with $0.52°C$ for OSTIA), and EMD for GOES-16 is higher than that for OSTIA, at every location except the floating lidars (an average of 0.27, compared with 0.21 for OSTIA).

Although OSTIA on average, across all buoys surveyed, better represents measured SSTs, GOES-16 performs better on average across only the floating lidar sites—Atlantic Shores, NYSERDA E05, and NYSERDA E06 (Table 6).

**Table 5.** Validation metrics for each remotely sensed data source at Atlantic Shores, NYSERDA E05, and NYSERDA E06 on a 10-min. output interval for July 2020.

| | Data Source | Bias (°C) | RMSE (°C) | $R^2$ | EMD |
|---|---|---|---|---|---|
| **Atlantic Shores** | MUR25 | 0.41 | 0.90 | 0.68 | 0.49 |
| | OSTIA | -0.17 | 0.71 | 0.8 | 0.37 |
| | GOES-16 | -0.01 | 0.76 | 0.76 | 0.35 |
| **NYSERDA E05** | MUR25 | 0.06 | 0.80 | 0.76 | 0.37 |
| | OSTIA | -0.14 | 0.59 | 0.88 | 0.32 |
| | GOES-16 | -0.13 | 0.63 | 0.86 | 0.31 |
| **NYSERDA E06** | MUR25 | 0.21 | 0.83 | 0.68 | 0.28 |
| | OSTIA | -0.14 | 0.61 | 0.83 | 0.27 |
| | GOES-16 | -0.02 | 0.64 | 0.81 | 0.2 |

**Table 6.** Average performance metrics for OSTIA and GOES-16 SSTs across the three floating lidar sites for June and July 2020.

| Metric | OSTIA | GOES-16 |
|---|---|---|
| **Bias (°C)** | -0.21 | -0.15 |
| **RMSE (°C)** | 0.67 | 0.65 |
| **Correlation** | 0.97 | 0.97 |
| **EMD** | 0.31 | 0.27 |

## 3.2 Monthly Wind Speeds

The probability distribution functions (PDFs) of hub-height wind speeds at each lidar show that WRF, on a monthly time scale and with a 10-minute output resolution, generally captures the shape of the observed wind speed distribution at each lidar (Fig. 4), which suggests that the model itself is performing as it should. The wind speed distributions for each simulation maintain an even closer similarity in shape to one another, which highlights the biases directly related to the particular SST data set being used. A box plot of wind speeds across the entire domain for each simulation and for both months (not shown) indicates that although GOES-16 and OSTIA present near-identical average hub-height wind speeds, GOES-16 winds show a greater spread than OSTIA. Additionally, in both simulations, whole-domain winds in June tend to be significantly faster than July winds (Fig. 4).

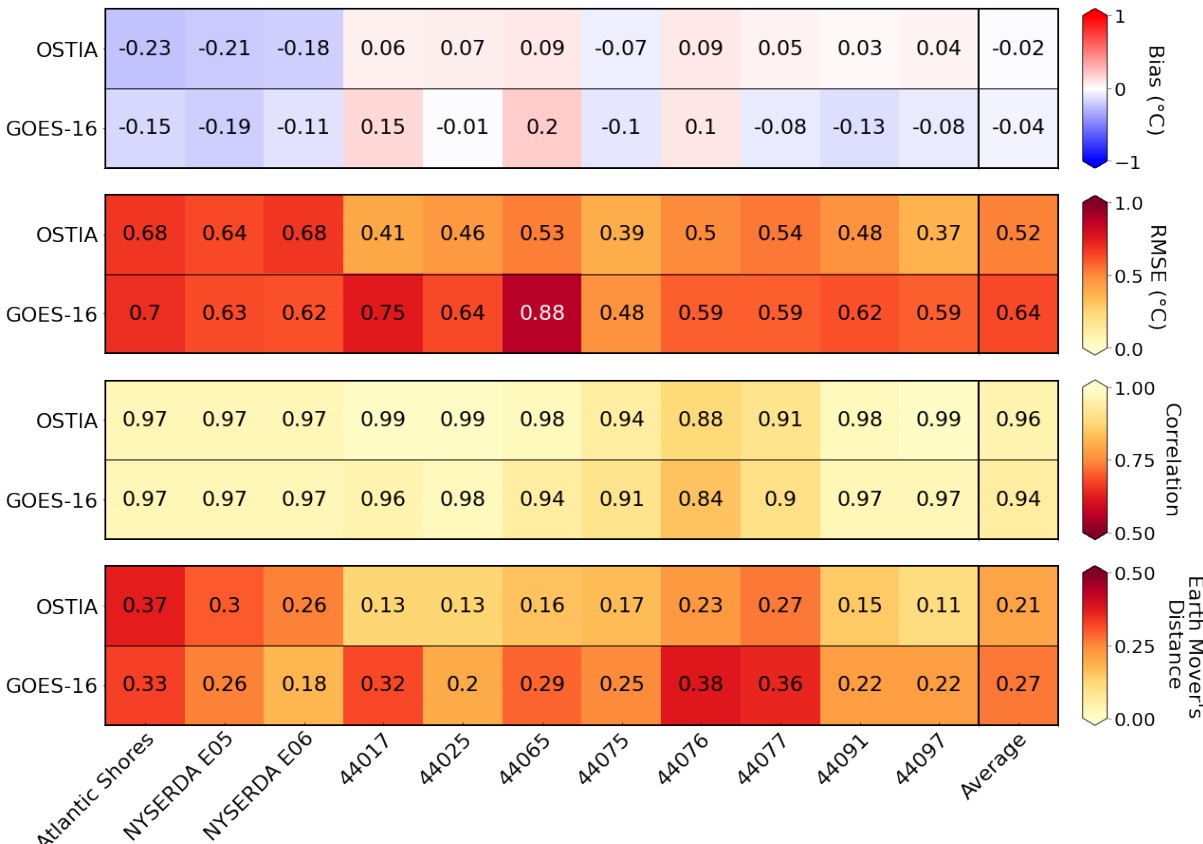

**Figure 3.** Mean bias, RMSE, correlation, and EMD for GOES-16 and OSTIA SSTs at each buoy location shown, combined for June and July, along with the average metrics over all sites for each product.

A comparison between each of the two simulations of monthly hub-height wind speed bias and correlation shows that they perform quite similarly during both June and July (bias and correlation shown in Fig. 5). Both products over-predict wind speeds, except at the two NYSERDA lidars during June. The correlation of each product's forecasted winds with observations, from 100 m ASL and higher, is above 0.65 at all three lidars during both June and July.

A map of the domain displaying the June and July average wind speed differences between the GOES-16 and OSTIA-forced simulations, overlaid by wind barbs indicating the average GOES-16 wind speeds, is shown in Fig. 6. In general, wind speeds deviate from each other only by small amounts on a monthly time scale. The results show maximum average wind speed differences between the two simulations of up to 0.25 m s$^{-1}$ for each month.

## 3.3 Event-Scale Wind Speeds

Using the event selection algorithm detailed in Section 7, we locate three periods—one in June and two in July—during which 140-m wind speeds in the OSTIA and GOES-16 simulations differ from one another by statistically significant amounts and

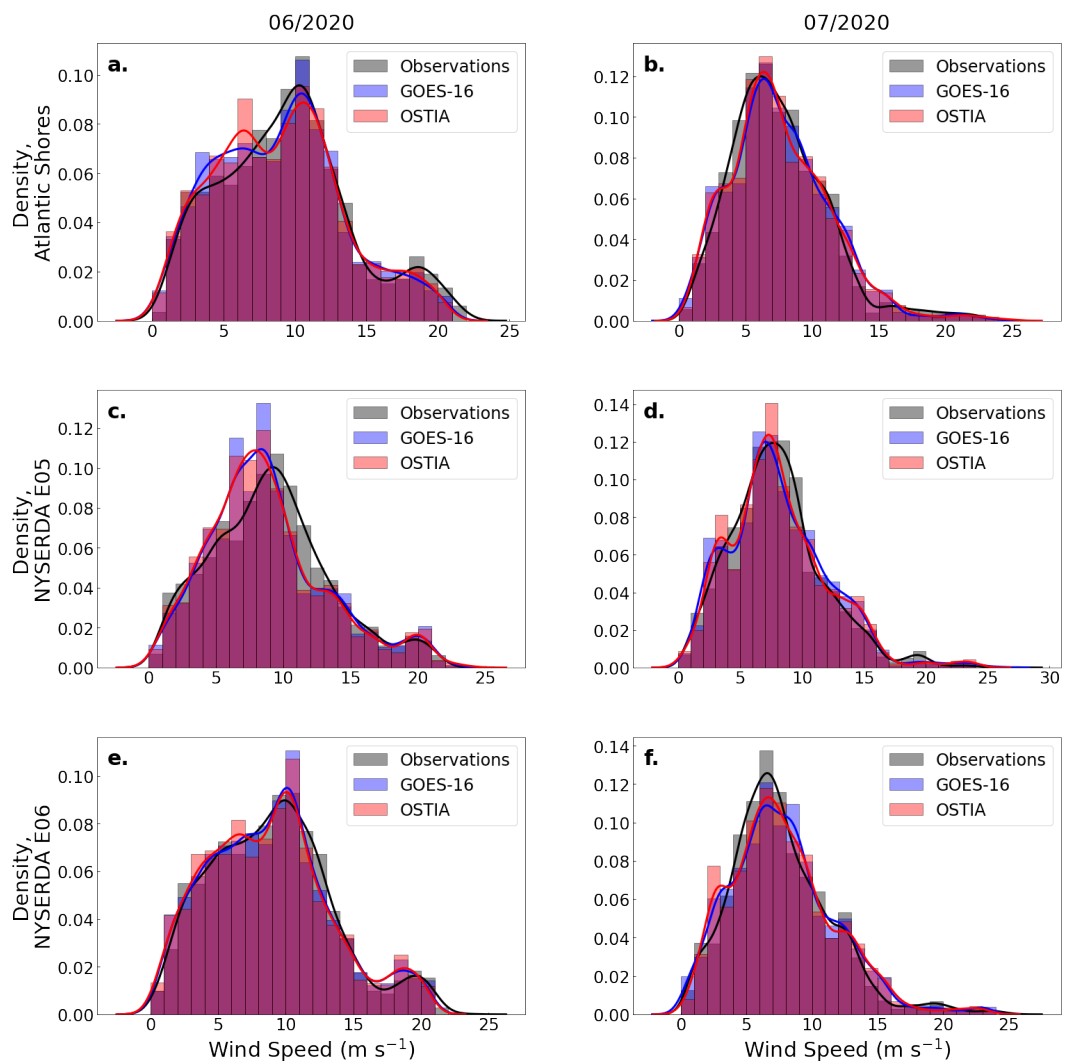

**Figure 4.** PDFs of 140-m wind speeds at each lidar location, for June (left) and July (right), taken via observations (gray), GOES-16-forced model output (blue), and OSTIA-forced model output (red).

still both validate relatively well against observations ($r^2 > 0.6$ at 2 or more lidars), per the criteria outlined in Section 7. These events are listed in Table 7. Lidars at which there are relatively large observational data gaps during these time periods (> 20%) are not considered. Therefore, data from only two locations for the June event and the second July event are used. For all three cases, at each lidar being considered, GOES-16 delivers overall stronger characterization of 140-m wind speeds than OSTIA. Validation metrics vary at different heights, so vertical profiles of bias, $r^2$, RMSE, and EMD for each event can be found in Appendix A.

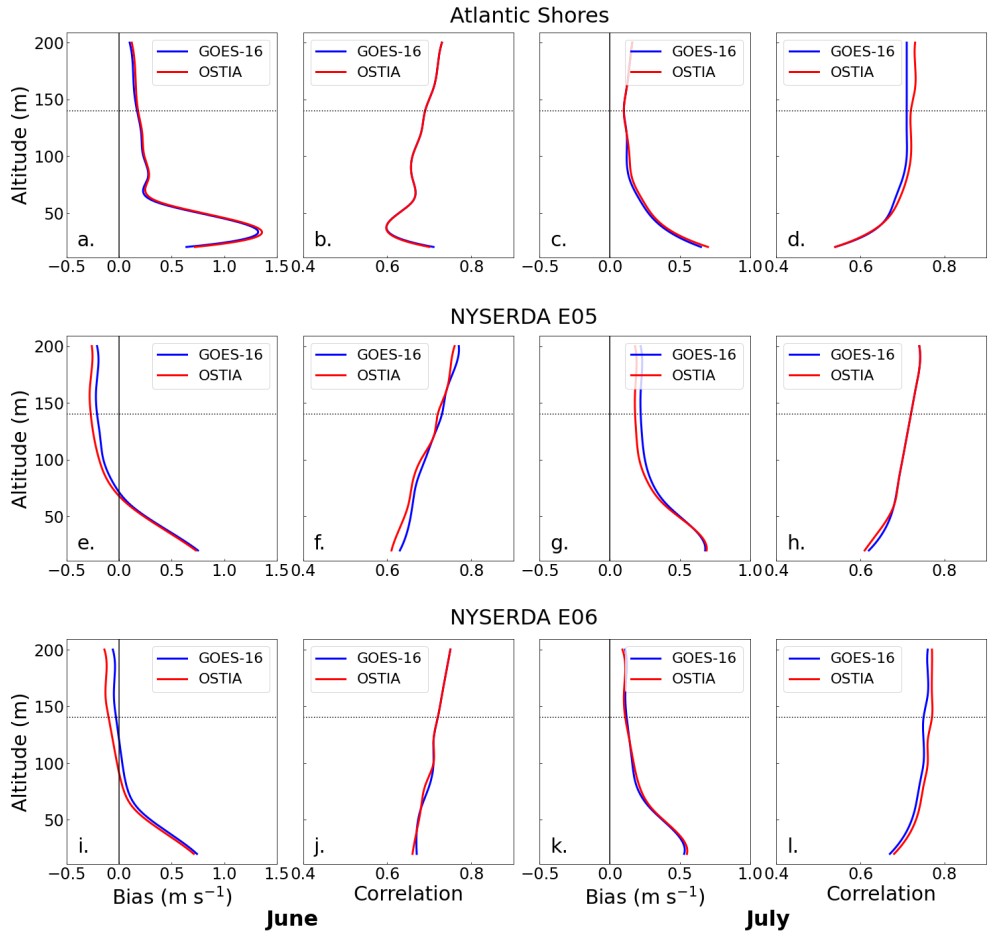

**Figure 5.** Modeled hub-height wind speed bias and correlation for simulations forced with GOES-16 (blue) and OSTIA (red) SSTs during June and July 2020 at the Atlantic Shores lidar (a, b, c, d), the NYSERDA E05 lidar (e, f, g, h), and the NYSERDA E06 lidar (i, j, k, l).

**Table 7.** Weather events selected for evaluation.

| Start Date | End Date | Event Length |
|---|---|---|
| 06-21-2020 13:40:00 | 06-22-2020 15:20:00 | 25h 40m |
| 07-10-2020 06:25:00 | 07-11-2020 09:00:00 | 26h 25m |
| 07-18-2020 02:05:00 | 07-18-2020 14:10:00 | 12h 5m |

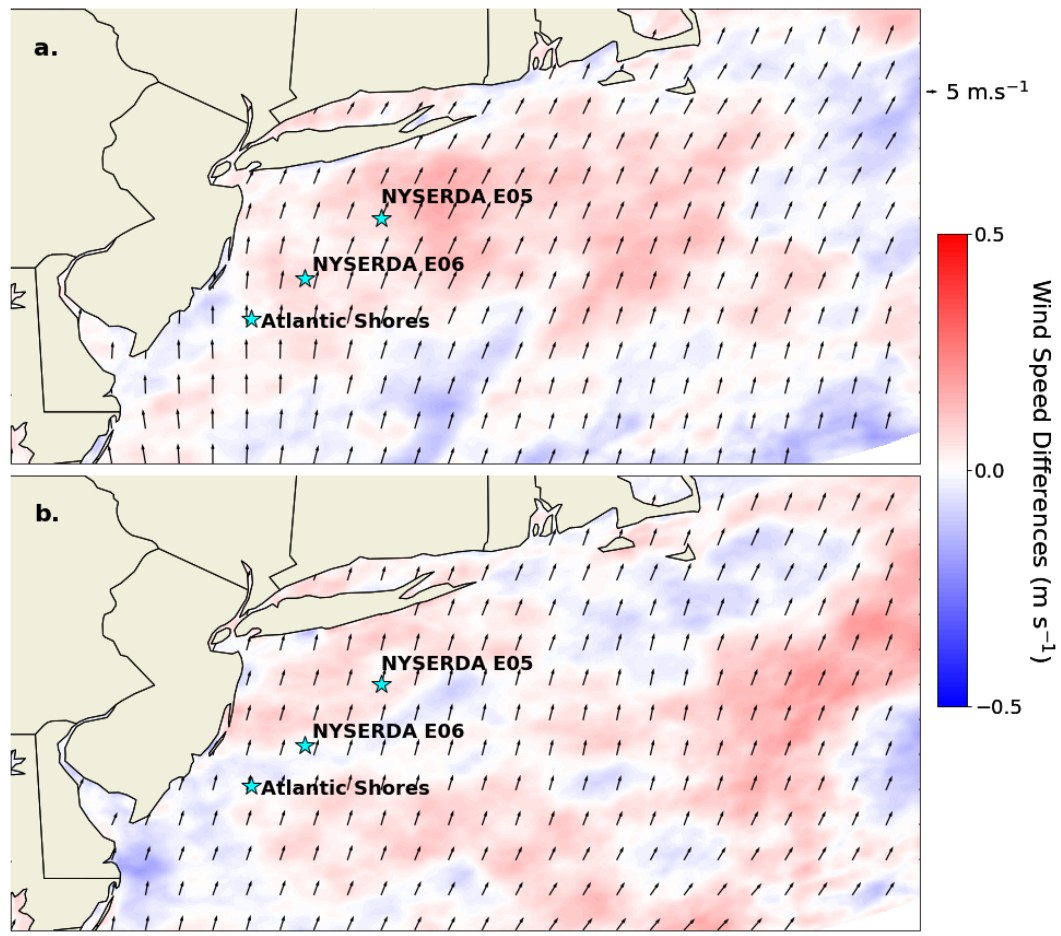

**Figure 6.** Modeled hub-height (140-m) wind speed differences, GOES-16 - OSTIA, for June (a.) and July (b.).

### 3.3.1 June 21-22, 2020 Event

We have identified an event that meets our criteria beginning on 06-21-2020 at 13:40:00 and ending on 06-22-2020 15:20:00. During this time period, offshore winds near the coast are southerly, with a tendency to follow the coastline as they rotate around a high-pressure system southeast of New Jersey.

Differences in SSTs between the two simulations do not correspond with significant 140-m wind speed differences at the three lidars; however, in other areas of the domain, including a region planned for development just south of Rhode Island (Fig. 1, marked by "RI"), these differences are larger, with magnitudes reaching over 2 m s$^{-1}$ (Fig. 7). There are not significant SST differences noted within that the region, although land surface temperatures may contribute to a stronger temperature gradient.

A planar depiction of 140-m wind speeds across the entire domain shows that, although the mean of the magnitude of the differences is roughly 0.25 m s$^{-1}$, they vary by location across the region. The maximum difference in average wind speeds

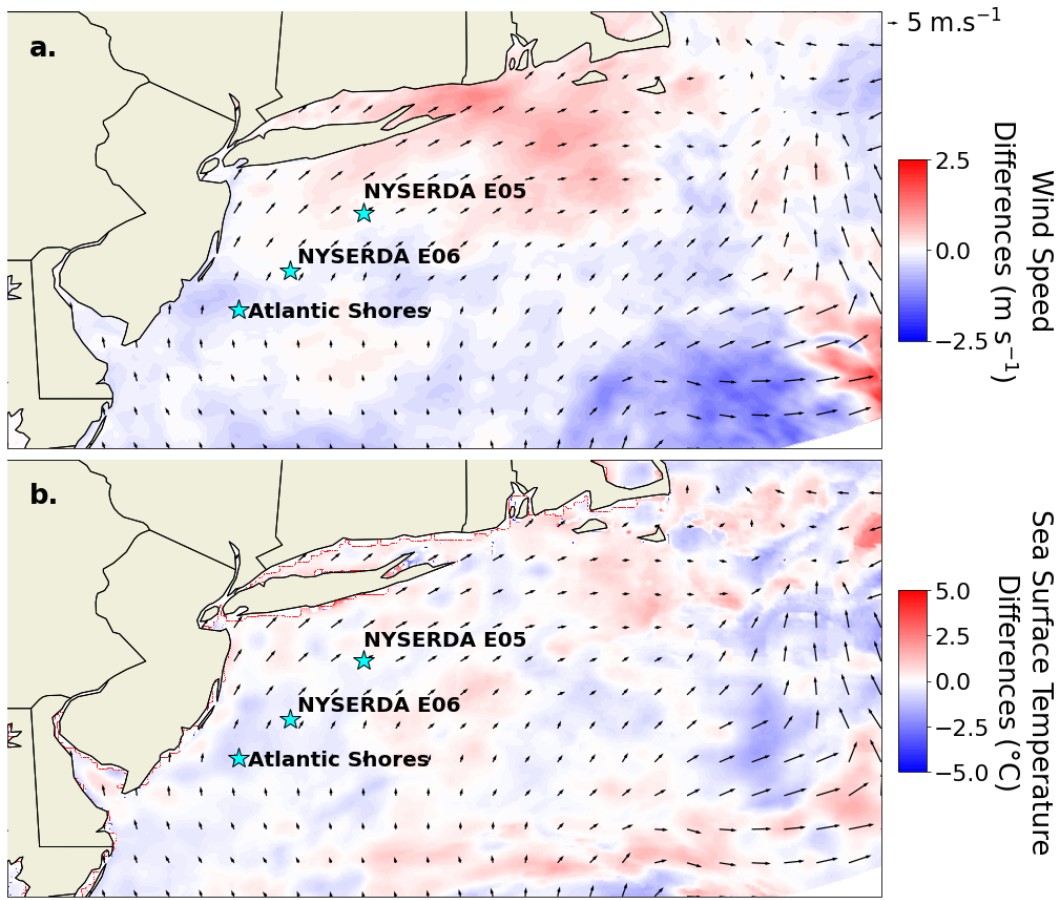

**Figure 7.** Differences (GOES-16 - OSTIA) in average 140-m wind speeds (a.) and SSTs (b.) over the entire domain for the June 21-22 event.

during this event is actually 2.25 m s$^{-1}$. This is noteworthy, as wind speed differences of this size have significant implications regarding power generation.

Model output from the two simulations captures the observed trend of a wind speed increase—but model outputted wind
speeds deviate significantly from each another leading into a first ramping event, during a second ramp event, and once wind speeds began to stabilize (Fig. 8). Observational data are only available at the two NYSERDA lidars during these periods. The Atlantic Shores lidar has significant data gaps, so validation at this location is not conducted.

Event validation metrics at both lidars indicate stronger performance by the GOES-16 simulation than OSTIA, particularly at vertical levels associated with typical offshore turbine hub heights. Validation metrics at hub height (140 m) for the two
simulations at NYSERDA E05 and NYSERDA E06 are shown in Fig. 9. Averaged correlation between the two sites is 0.78 for GOES-16 and 0.76 for OSTIA. The bias at each site is comparable between the two simulations, at -0.26 m s$^{-1}$ and -0.24 m s$^{-1}$ for OSTIA and GOES-16, respectively. Model performance across the vertical should be taken into consideration if

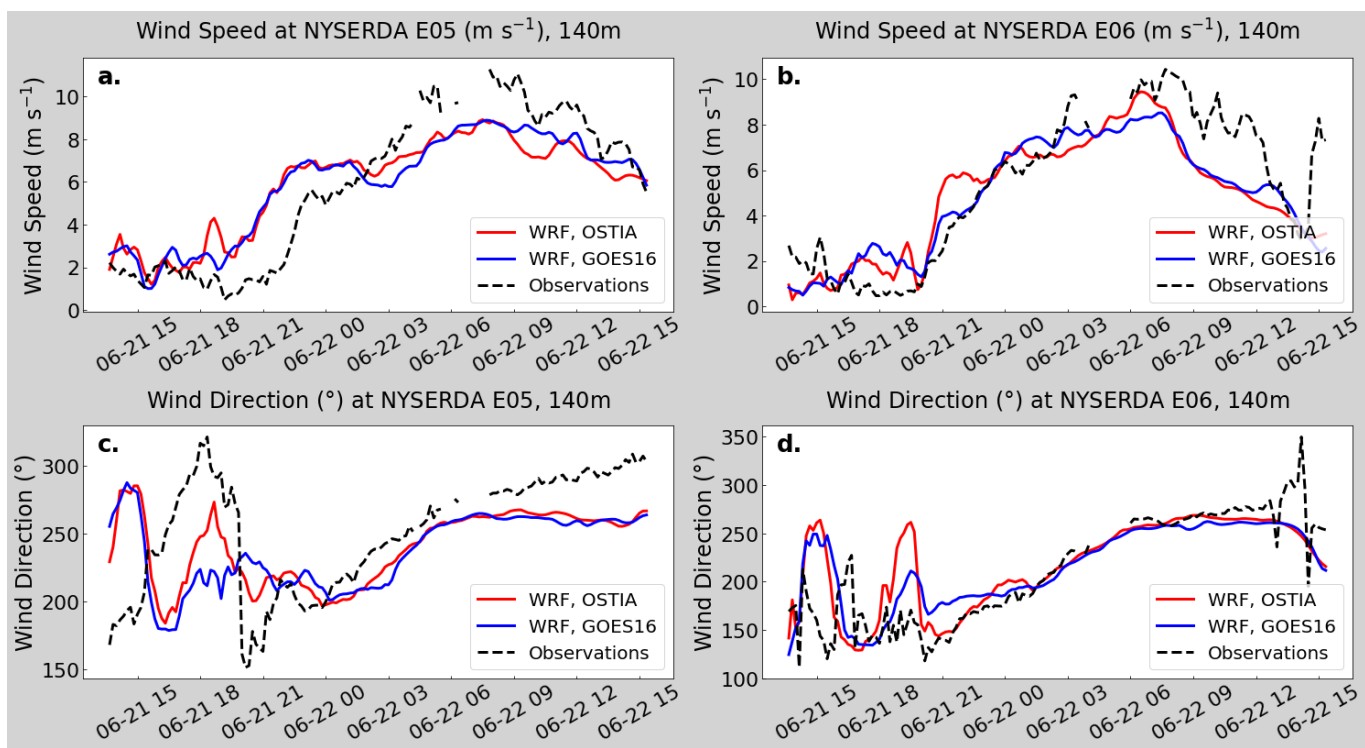

**Figure 8.** Hub-height (140-m) wind speeds and wind directions at the NYSERDA E05 (a, c) and NYSERDA E06 (b, d) lidars during the June 21-22 event. Atlantic Shores is not shown due to a lack of observational data.

the rotor-equivalent wind speed is used to calculate power generation during extreme shear events as well as when looking at turbines of varying hub heights. Of note is that both RMSE and EMD are lower for GOES-16 at all heights (Appendix A).

### 3.3.2 July 10-11, 2020 Event

We have next flagged an event that occurred between 07-10-2020 06:25:00 and 07-11-2020 09:00:00. Synoptically, Tropical Storm Fay was observed to be moving in a southerly direction through the region during this time. High wind speeds, which peaked at 07-10-2020 18:00:00, may be attributed to this storm. Average wind directions also reflect the storm path (Fig. 10). Differences in average wind speed between the two simulations peak at only 1.35 m s$^{-1}$—which is around 1 m s$^{-1}$ less than the maximum difference during the June event. The mean difference in the magnitudes of average wind speeds across the domain is 0.2 m s$^{-1}$.

With the winds blowing in a south-southeasterly direction across the leasing area, and the differences in GOES-16 showing a relatively stronger temperature gradient across that area in the direction of the wind, the GOES-16 simulation outputs stronger wind speeds compared with OSTIA (Fig. 10). This is reflected in the overall bias seen at the three lidars, although both simulations overall underestimate wind speeds during this time (Fig. 11).

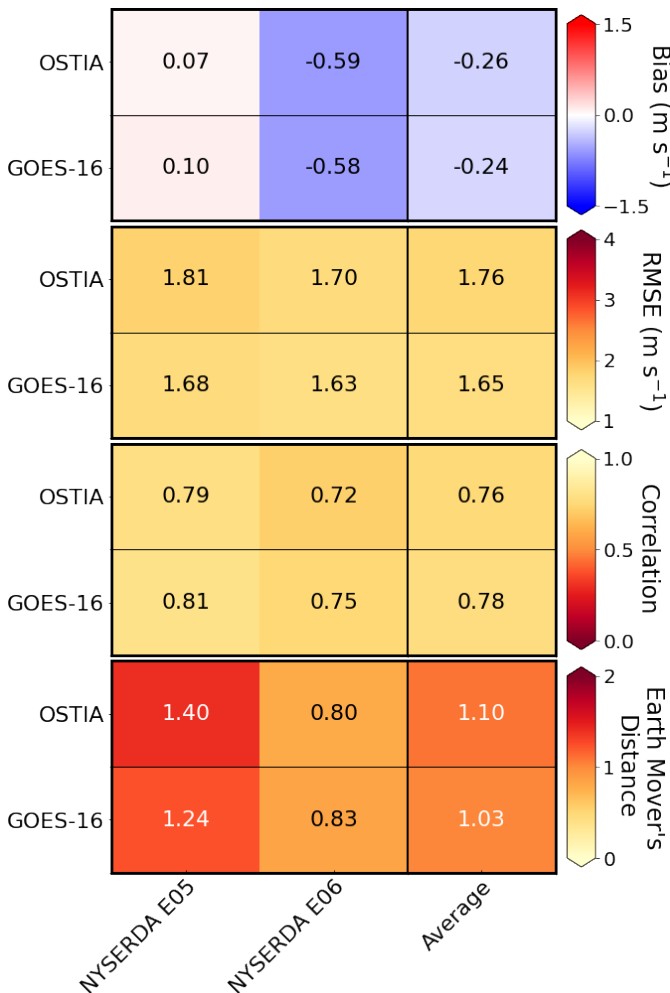

**Figure 9.** Hub-height wind speed validation metrics at the NYSERDA E05 and NYSERDA E06 lidars during the June 21-22 event: bias, RMSE, correlation, and EMD.

Near-complete wind data sets exist at all three lidars for the duration of this event. Validation metrics show a correlation of 0.45 or greater between model output and observations, for both simulations, at 140 m at each lidar site. The models perform the best, across all four metrics, at NYSERDA E06. Average $r^2$ between the two products is comparable, at 0.62. Bias strength peaks at -1.29 m s$^{-1}$ at Atlantic Shores in the GOES-16 simulation. The bias is strongly negative at Atlantic Shores, but is positive at NYSERDA E05, the northernmost lidar, in the GOES-16 simulation. On average, bias in the OSTIA simulation is 0.11 m s$^{-1}$ stronger (more negative) than that in the GOES-16 simulation. Hub-height (140-m) validation metrics for each simulation at each lidar are shown in Fig. 11.

The GOES-16 and OSTIA simulations output statistically significantly different 140-m wind speeds during this period. Both generally track the major observed wind speed patterns at each observation site, although the timing and magnitude of some

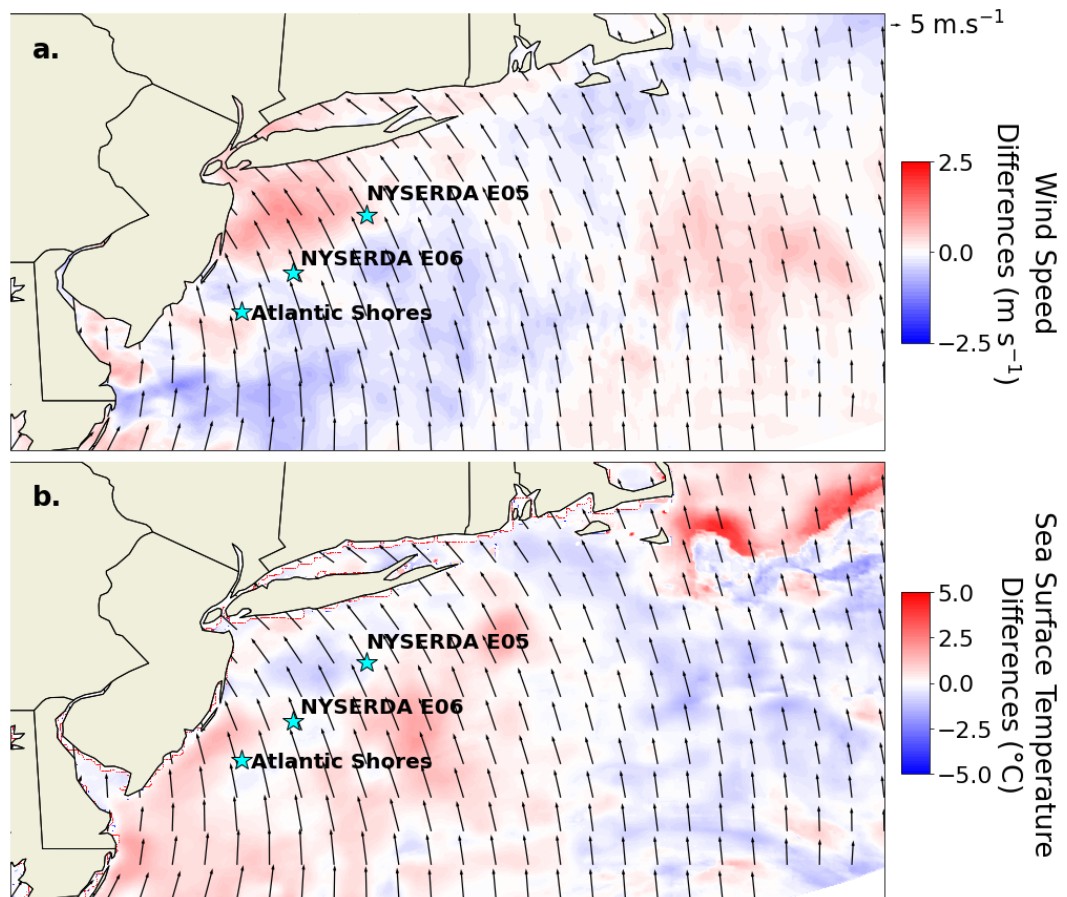

**Figure 10.** Differences (GOES-16 - OSTIA) in average 140-m wind speed (a.) and average SST (b.) during the July 10-11 event.

longer-scale changes is missed by the models. For example, although the models captured the wind speed increase beginning at 07-10-2020 09:00:00 at Atlantic Shores, both erroneously forecast a subsequent down-, up-, and then down-ramp event (Fig. 12(a.)). At NYSERDA E05, the initial spike in observed speeds occurred several hours earlier than the models forecasted (Fig. 12(c)). And, at NYSERDA E06, the models miss the timing and magnitude of a dip in wind speeds that was observed between 07-10-2020 15:00:00 and 07-10-2020 20:00:00; they forecast a much larger drop in speed, almost 10 m s$^{-1}$ compared with the original 6 m s$^{-1}$, beginning just as the observed dip recovers. These faults all contribute to the relatively lower correlation of model output from both simulations with observations for this event.

### 3.3.3 July 18, 2020 Event

The last flagged event occurred between 07-18-2020 02:05:00 and 07-18-2020 14:10:00. During this time, a high pressure system was moving westerly into the region, causing a fall in wind speeds throughout the morning. This front spanned eastward into the Atlantic and aligned perpendicular to the New Jersey coast. Hub-height wind speed differences between the two

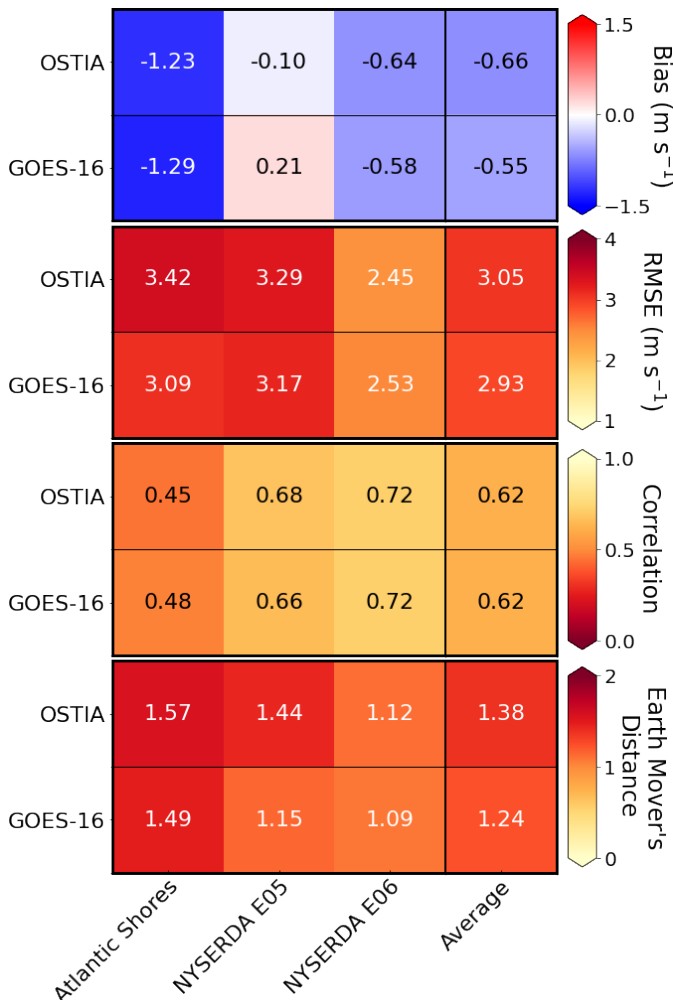

**Figure 11.** Wind speed validation metrics at an 140-m hub height at the Atlantic Shores, NYSERDA E05, and NYSERDA E06 lidars during the July 10-11 event: bias, RMSE, correlation, and EMD.

simulations are varied throughout the domain, with a maximum difference in the magnitudes of average event wind speeds of 1.7 m s$^{-1}$, and a mean difference of 0.29 m s$^{-1}$. (Fig. 13). Larger differences, in particular, are found near the coast, and positive/negative differences vary spatially.

Sufficient data for this event are present at Atlantic Shores and NYSERDA E06, but not at NYSERDA E05. Validation analysis for the event at the two lidars indicates that overall the products perform comparably, although OSTIA shows an almost 0.5 m s$^{-1}$ stronger bias at NYSERDA E06. At lower heights (Fig. A3), there there is clear evidence of GOES-16 outperforming OSTIA at Atlantic Shores. Farther north, at NYSERDA E06, the relative performance of each simulation is variable. Validation metrics at Atlantic Shores and NYSERDA E06 are shown in Fig. 14. Of note is that PDFs of 140-m wind

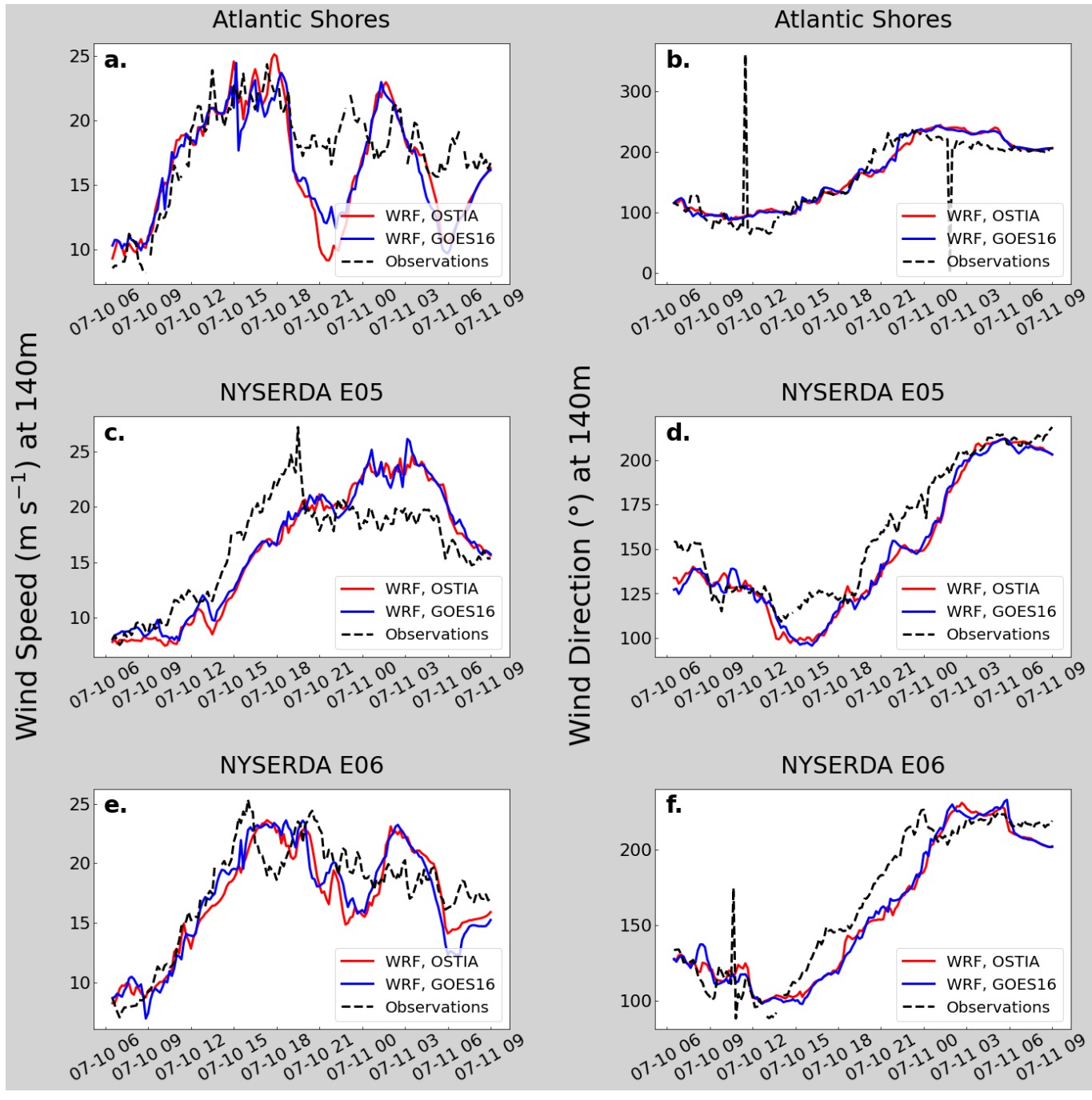

**Figure 12.** Hub-height (140-m) wind speed and wind direction at the Atlantic Shores lidar (a, b) the NYSERDA E05 lidar (c, d), and the NYSERDA E06 lidar (e, f) during the July 10-11 event.

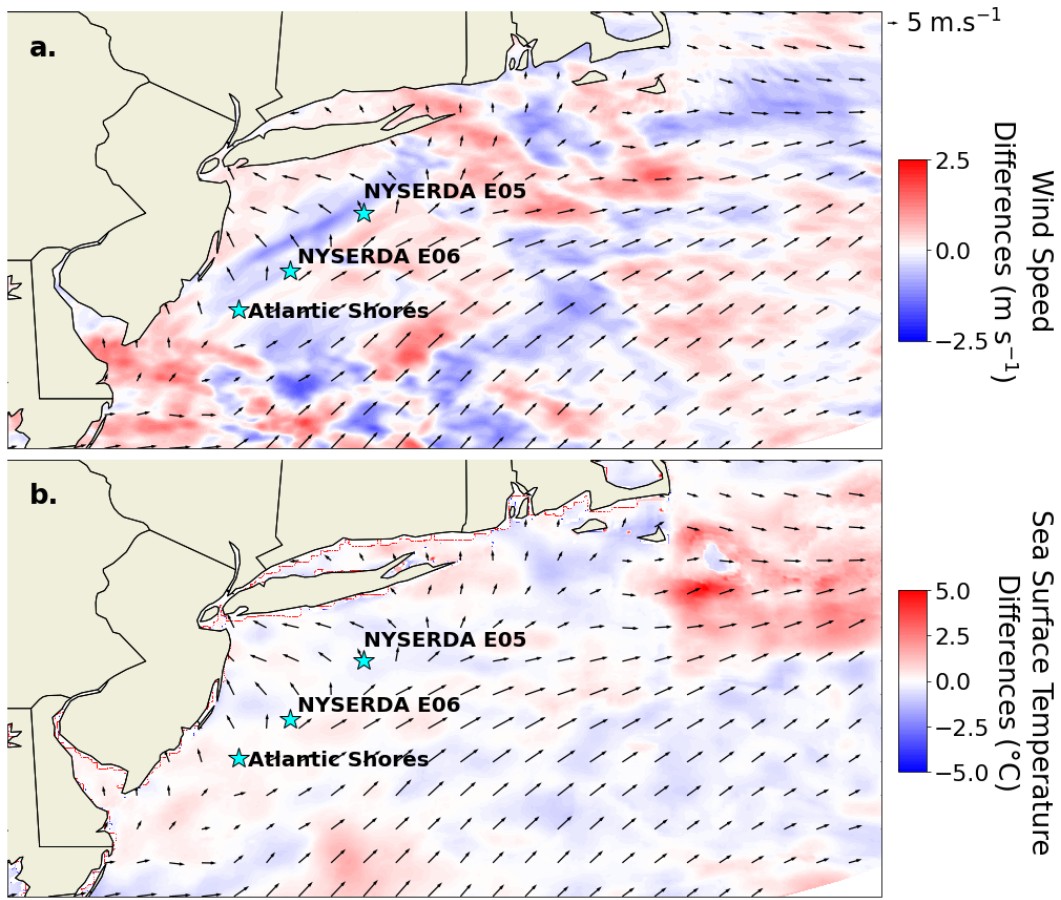

**Figure 13.** Differences (GOES-16 - OSTIA) in average 140-m wind speeds (a.) and average SSTs (b.) and during the July 18 event.

speeds during this time (not shown) indicate bimodality in the observed wind speed distribution. Although OSTIA captures this feature, GOES-16 misses it almost entirely—at both lidars.

Over this event's time period, 140-m wind speeds drop at both locations from over 14 m s$^{-1}$ to less than 3 m s$^{-1}$, before beginning to increase again. As with the previous two cases, the models capture the general wind speed trend of slowing throughout the duration of the event, although they both present errors in the magnitude of wind speeds and the timing of wind profile changes. In this case, at NYSERDA E06 they predict a shift in wind direction before one was observed (there is insufficient wind direction data at NYSERDA E05 to make a determination about if this holds true at both lidars). Both simulations exhibit greater spread in wind speeds at each lidar than observations, and both have slower average velocities (Fig. 15).

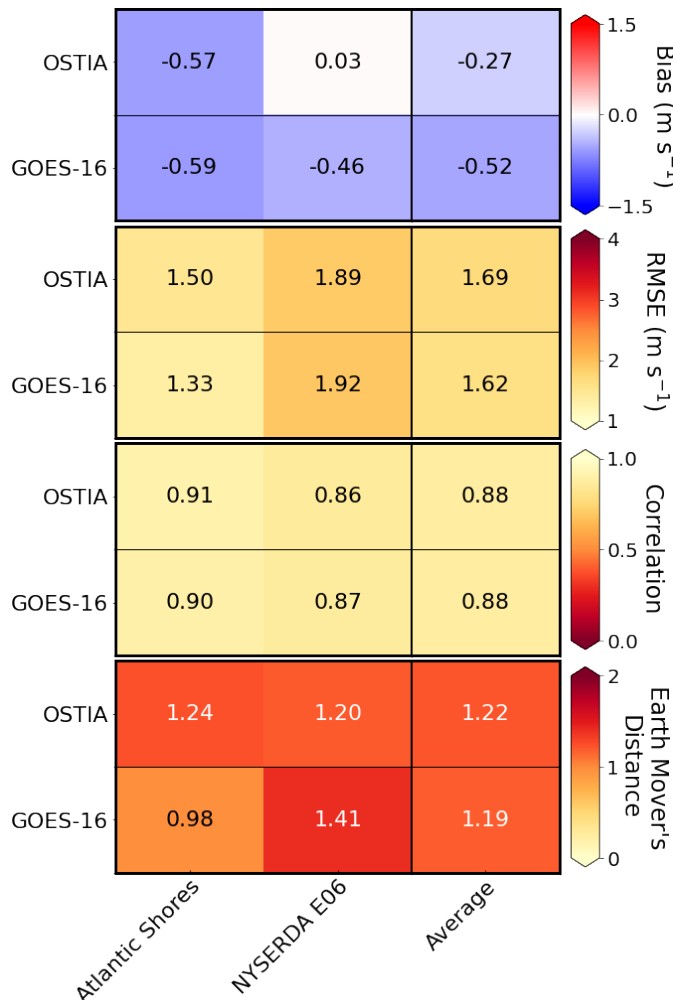

**Figure 14.** Hub-height (140-m) wind speed validation metrics at the Atlantic Shores and NYSERDA E06 lidars during the July 18 event: bias, RMSE, correlation, and EMD.

## 4   Discussion

In Section 3, we saw varied performance between the two SST products, depending on the variable in consideration (SST or winds) and the time scale (monthly or event-scale). SST in OSTIA validated better than GOES-16 across the buoy array. However, hub-height wind speeds at each lidar location, for each month, point to a similar performance by both data sets. More nuanced events that occurred over hourly-to-daily time scales, which were selected based on differences between simulation output and overall WRF performance, also indicate comparable performance between the two simulations, with, in general, a slightly stronger performance using GOES-16 SSTs. Events such as these are of importance to wind energy forecasts because we found that they often correlate with wind ramps, during which times wind speeds fall below the rated power section of

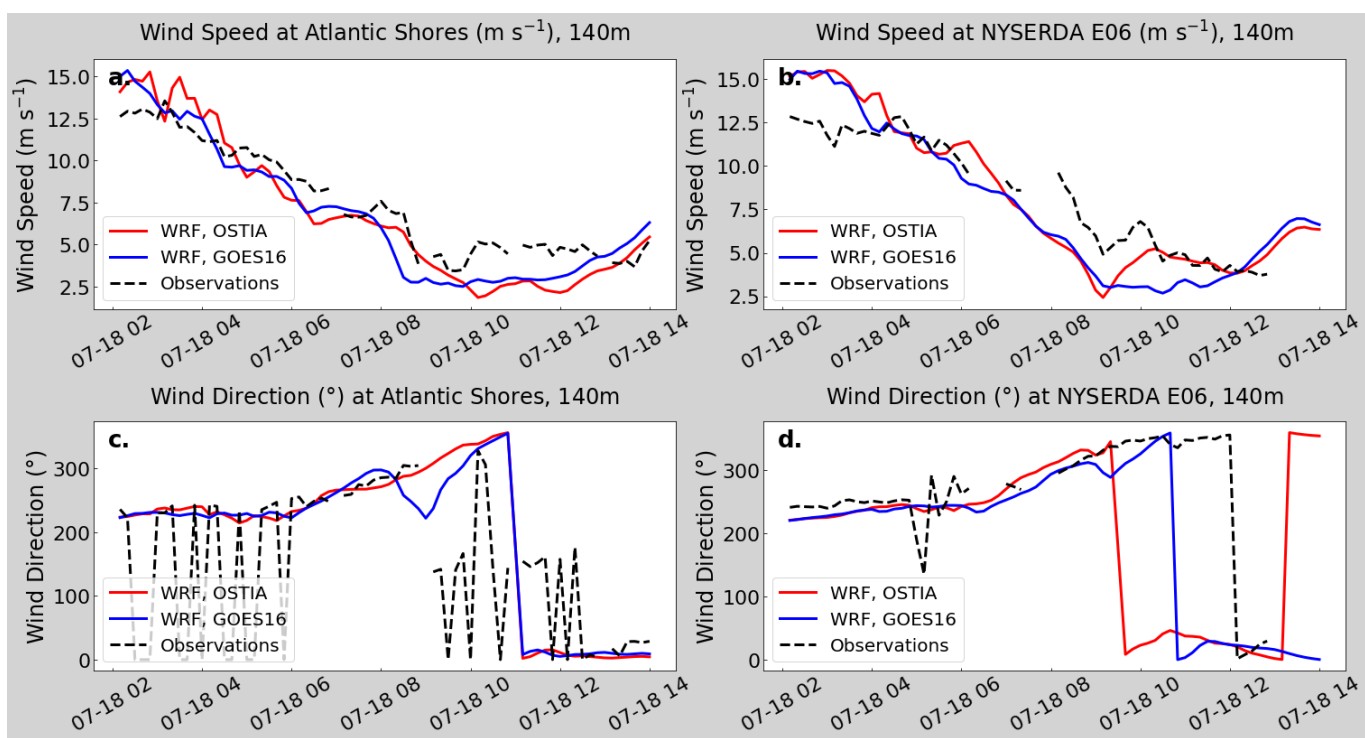

**Figure 15.** Hub-height (140-m) wind speed and wind direction at the Atlantic Shores lidar (a, c) and the NYSERDA E06 lidar (b, d) during the July 18 event.

turbine power curves. In this region, power output is very sensitive to fluctuations in wind speed. Therefore, obtaining the most accurate wind forecast within this regime is important.

Although OSTIA SSTs at the lidars validate better than GOES-16, modeled hub-height (140-m) winds do not demonstrate the same trend. Compared with the 140-m winds heat maps for the events (Fig. 9, 11, 14), good wind characterization perfor-
mance does not necessarily correspond to good SST validation (Appendix B), which implies that SSTs across the full study area, not just localized temperatures, may have an impact on winds in the leasing area.

To further examine the disjointedness between SST and wind speed point validations, we consider model performance during each event in conjunction with how well their corresponding SSTs validate with observations across the buoy array (where data are available). During the June 21-22 event, 140-m wind speed bias and EMD in both simulations are the lowest of all the cases
considered (Fig. 9), but there is an overall cold SST bias (-0.31°C for OSTIA, -0.84° for GOES-16) across the buoy arrays, and SST correlation is very poor (Fig. B1). During the July 10-11 event, OSTIA SST's validate better than GOES-16 (Fig. B2), but the two simulation's hub-height wind speeds validate similarly, with relatively poor performance across all validation metrics compared with the other flagged events. And, during the July 18 event, while SSTs from both products validated the best out of the three events (Fig. B3) and the models captured the overall trend in wind speeds with correlations of 0.88 for
both simulations, EMD and the magnitude of the bias were both greater than the June 21-22 event.

Across all three events there is no clear relationship between wind forecasting ability and SST correctness. However, as with the montly analysis, the relatively larger differences between SST and wind speed validations in the GOES-16 event simulations compared with those of OSTIA suggests that GOES-16's finer temporal and spatial resolutions may more accurately influence larger-scale dynamics throughout the region, leading to greater forecasting skill. Though beyond the scope of this study, more widespread in situ observations and validation of the two data sets against observations would help address how large of an effect this is. Such a study would additionally allow for a better understanding of how much a nuanced representation of diurnal SST cycling (as in the GOES-16 product, Fig. 2) influences the accuracy of wind forecasting in the region.

Rapid fluctuations in wind speed, on the order of minutes to hours, are overall not well captured by either simulation. This is most evident during the July 10-11 event, where the magnitude and timing of wind speed changes are most misaligned. Correlation in particular is low (0.62), and RMSE is high ( 3 m s$^{-1}$). In comparison, during the July 18 event, when wind speeds slow without any ramping, both simulations perform better; in particular, correlation is relatively strong for both (0.88). Similarly, during the June 21-22 event, which lacks any sharp ramping, correlation is also above 0.75 for both GOES-16 and OSTIA. Shorter, weaker fluctuations in wind speed during this event are not well captured by either simulation (Fig. 8), although GOES-16 validates with nominally less error (Fig. 9). It is possible that GOES-16's finer resolution may contribute to its slightly stronger performance in this case, as the product does capture hourly fluctuations in SSTs. In contrast, during the July 10-11 storm, winds are so strong and variable that SST resolution does not appear to make a significant difference, and the two simulations both show significant error.

Also of note are the differences in weather between June and July in the Mid-Atlantic region. June average wind speeds are faster than those in July for both simulations. Overall cooler SSTs throughout the region are also present in June, with a weaker temperature gradient in the cold bubble offshore of Cape Cod than in July. The difference in temperature in this region between GOES-16 and OSTIA is more pronounced in July, with GOES-16 showing warmer monthly average SSTs (Fig. 16). The corresponding average wind speed and surface pressure differences between the products are also more distinct in July. While the two simulations validated similarly for each month, in the more turbulent July environment, OSTIA outperforms GOES-16. However, due to the greater number of missing pixels that required gap-filling in July as compared with June (36.86% more over the course of each month, resulting from the increased cloud cover—and the possible stronger coastal cold pool), this indication of product superiority may be questionable. The addition of more refined post-processing to account for incorrectly cloud-filtered pixels, as shown in Murphy et al. (2021), has been demonstrated to improve the accuracy GOES-16 SSTs. A comparison of simulations using OSTIA against those using the sophisticated GOES-16 product should be conducted in the future to evaluate the level of improvement delivered using the latter's more computationally expensive data set.

## 5 Conclusions

In this study, we evaluated how SST inputs affect how well WRF forecasts winds off the Mid-Atlantic coast. We initially considered three remotely-sensed SST data sets and validated them against in-situ observations taken at various buoys in the region. The first data set, OSTIA, has a 0.05°resolution, is output at a daily interval, and is assimilated with in situ observations.

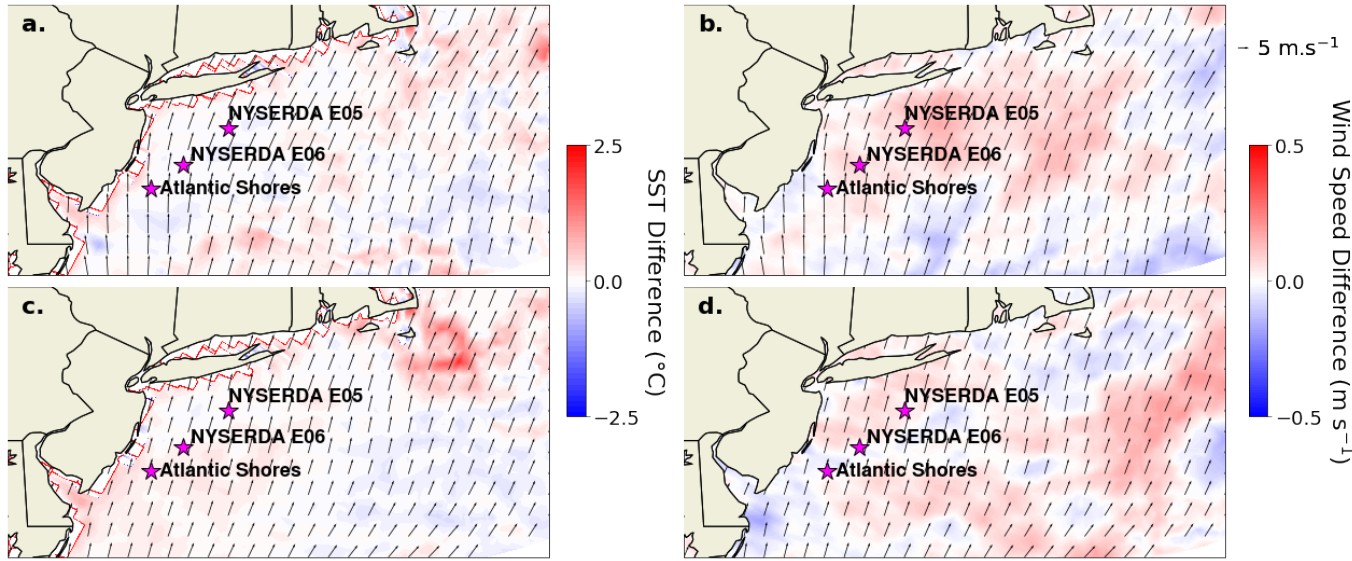

**Figure 16.** Differences (GOES-16 - OSTIA) in average SST for June (a.) and July (c.), and in modeled 140-m wind speeds for June (b.) and July (d.).

The second product, GOES-16, has a finer resolution of 0.02°and is generated on an hourly interval, but does not have the same
level of post-processing as OSTIA. The data set contains gaps and lacks assimilation with in situ observations. To account for missing data, we ran an EOF process to statistically analyze trends in each monthly data set and fill its gaps. The third SST data set, MUR25 has a 0.2° spatial resolution and a daily temporal resolution, and similar to OSTIA it is assimilated with in situ observations. MUR25 was removed from further analysis after SST validation, as its performance compared with the other two data sets was notably poor.

Following SST validation, WRF was run for June and July of 2020 in the Mid-Atlantic domain depicted in Fig. 1. The simulation setups were identical with the exception of the input SST field (key model parameters presented in Table 2). We compared the characterized 140-m (hub-height) winds against lidar observations by calculating correlation, bias, RMSE, and EMD on both monthly and event-length (hourly) timescales. Times during which there was significant difference in wind speed output from the two simulations and the general WRF performance was satisfactory were flagged as events. Our findings indicated that while OSTIA SSTs validated better against buoy measurements, the model validated comparably when forced with each SST data set, with GOES-16 having output nominally more accurate wind speeds. These findings suggest that the differences in temporal and spatial resolution between the two SST products influence wind speed characterization, with finer resolutions leading to better forecasting skill.

Overall, this study shows that SST inputs to WRF do affect forecasted winds and how well they validate against observations in future leasing areas in the Mid-Atlantic. Although GOES-16 SSTs validate worse against buoy observations than OSTIA SSTs, on both monthly and event timescales, the wind characterization produced using GOES-16 inputs validates as well

as, or in some cases better than, the OSTIA characterization, indicating that finer temporal and spatial SST resolution may contribute to improved wind forecasts in this region. To further explore this, a higher-quality post-processed GOES-16 data set assimilated with in situ observations, such as that created by Murphy et al. (2021), could be evaluated in future studies; generating a comparable data set was beyond the scope of this study.

This step of the research is in its early stages and more cases need to be considered—on both monthly and event scales. Future work should focus on identifying and analyzing more promising events—in particular, wind ramps and low pressure systems, as they have shown to produce larger differences between GOES-16 and OSTIA-forced model output. Wind direction validation, as well, should be considered in the future, as it was beyond the scope of this study.

 **Appendix A:  Average validation metrics through 200 m for flagged cases**

Contents of this Appendix include plots of horizontally averaged bias, correlation, RMSE, and EMD from sea level through a vertical height of 200 m for each flagged case, at each relevant lidar. Fig. A1 shows metrics at the two NYSERDA lidars for the June 21-22 event, Fig. A2 shows metrics at all three lidars for the July 10-11 event, and Fig. A3 shows metrics at Atlantic Shores and NYSERDA E06 for the July 18 event. The GOES-16 simulations are indicated by the blue lines and labeled with
385 "G", while the OSTIA simulations are shown in red and labeled with "O." The labels AS, E05, and E06 refer to Atlantic Shores, NYSERDA E05, and NYSERDA E06, respectively.

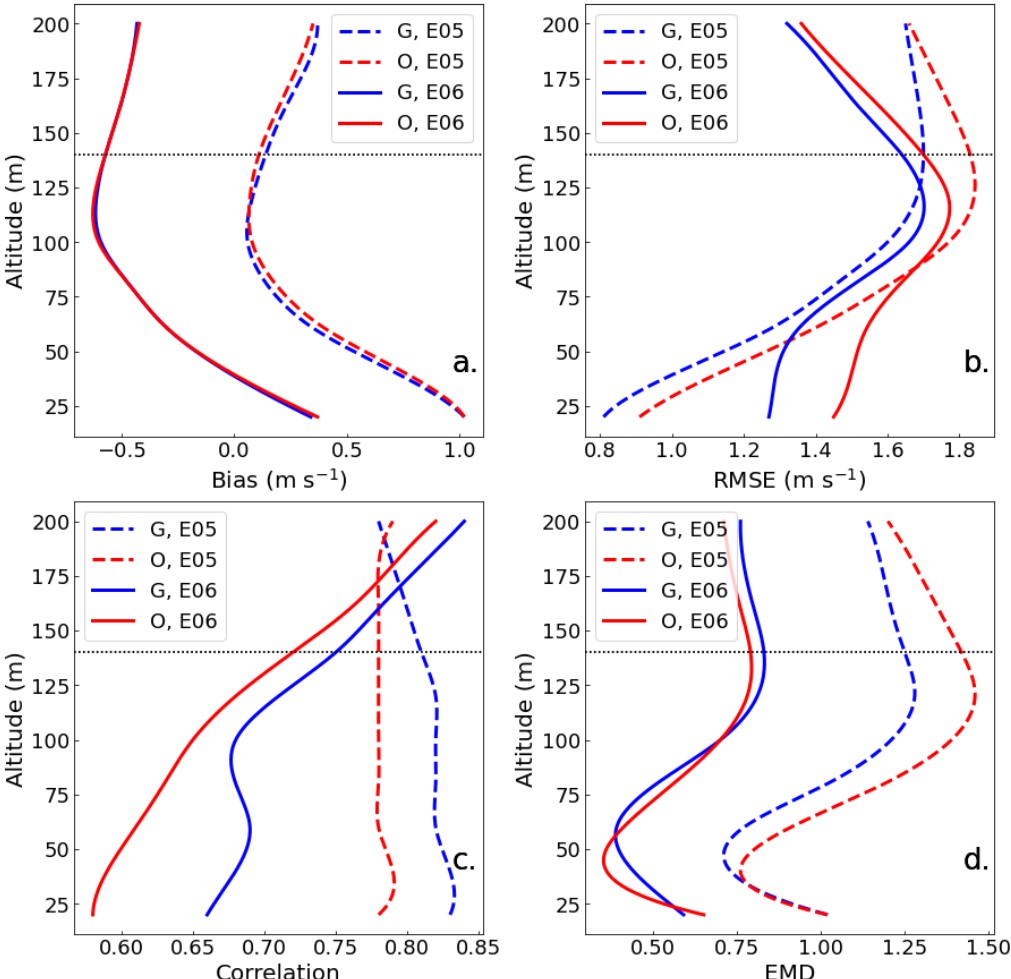

**Figure A1.** Validation metrics at NYSERDA E05 and NYSERDA E06 during the June 21-22 event: bias (a.), RMSE (b.), correlation (c.), and EMD (d.). Hub height is marked by a horizontal line at 140 m. The label "G" refers to GOES-16 simulations, while "O" refers to OSTIA simulations.

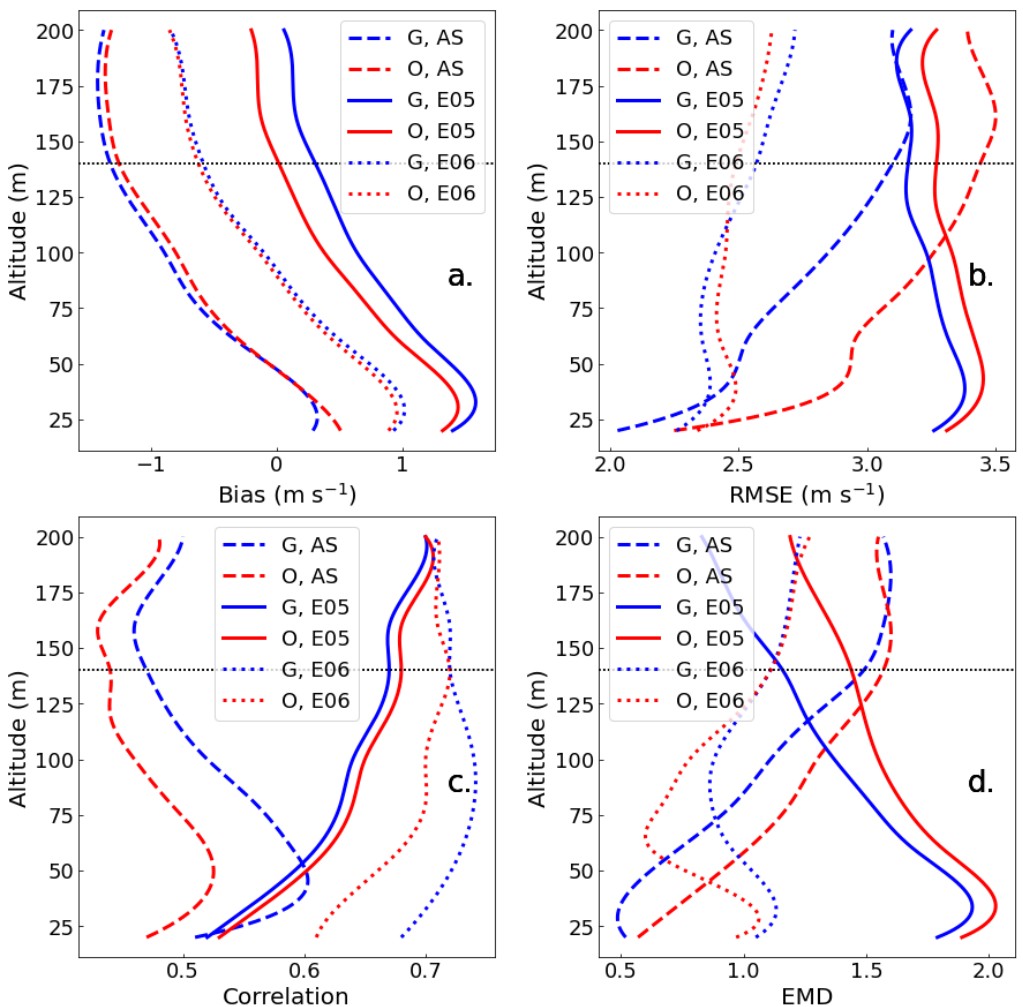

**Figure A2.** Validation metrics at Atlantic Shores, NYSERDA E05, and NYSERDA E06 during the July 10-11 event: bias (a.), RMSE (b.), correlation (c.), and EMD (d.). Hub height is marked by a horizontal line at 140 m. The label "G" refers to GOES-16 simulations, while "O" refers to OSTIA simulations.

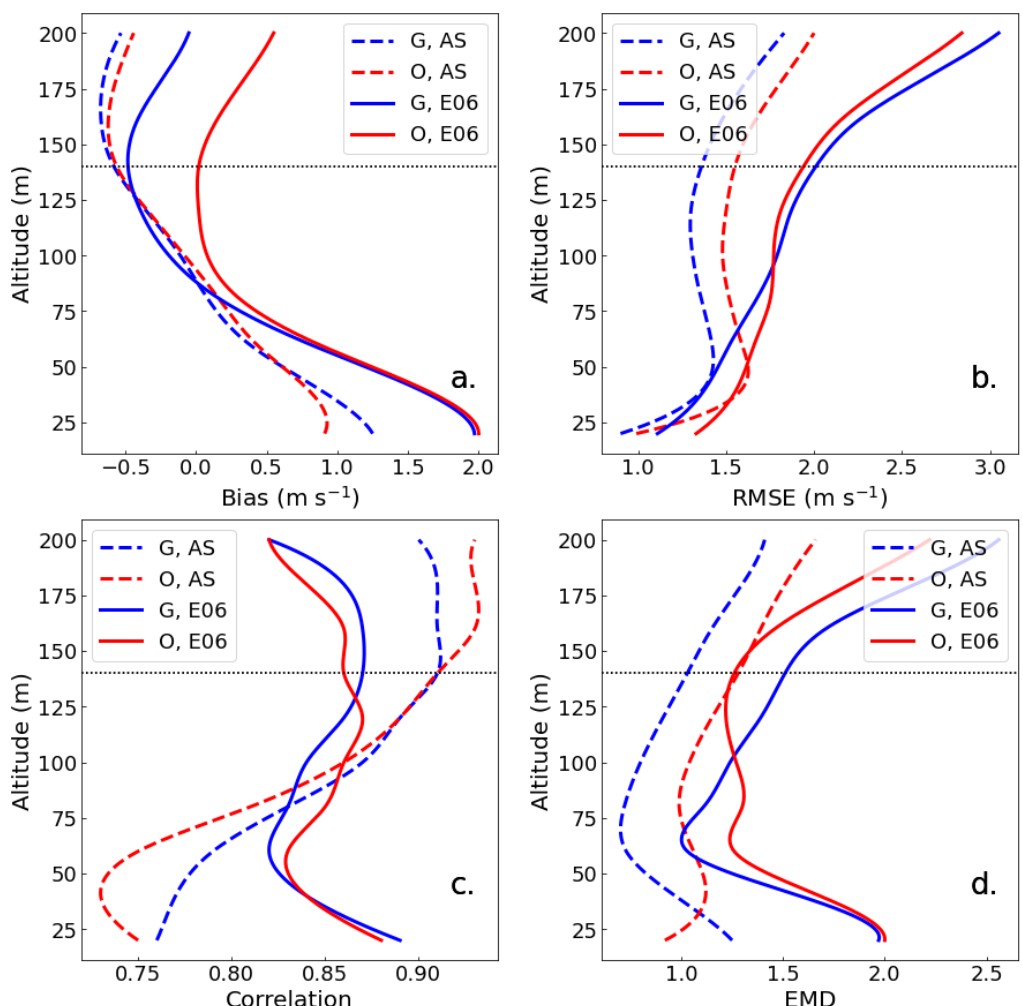

**Figure A3.** Validation metrics at Atlantic Shores and NYSERDA E06 during the July 18 event: bias (a.), RMSE (b.), correlation (c.), and EMD (d.). Hub height is marked by a horizontal line at 140 m. The label "G" refers to GOES-16 simulations, while "O" refers to OSTIA simulations.

## Appendix B:  Sea Surface Temperature Validation Metrics for the Flagged Events

Contents of this Appendix include heat maps of SST validation metrics for each of the three events flagged in this study. Each plot includes a different subset of the buoy array; locations with missing data were removed from analysis on a case by case basis. The average value for each metric across all buoys considered is located on the far right side of each row. The validation metrics considered are bias, RMSE, correlation, and EMD. Fig. B1 shows SST validation metrics for the June 21-22 event, Fig. B2 shows SST validation metrics for the July 10-11 event, and Fig. B3 shows SST validation metrics for the July 18 event.

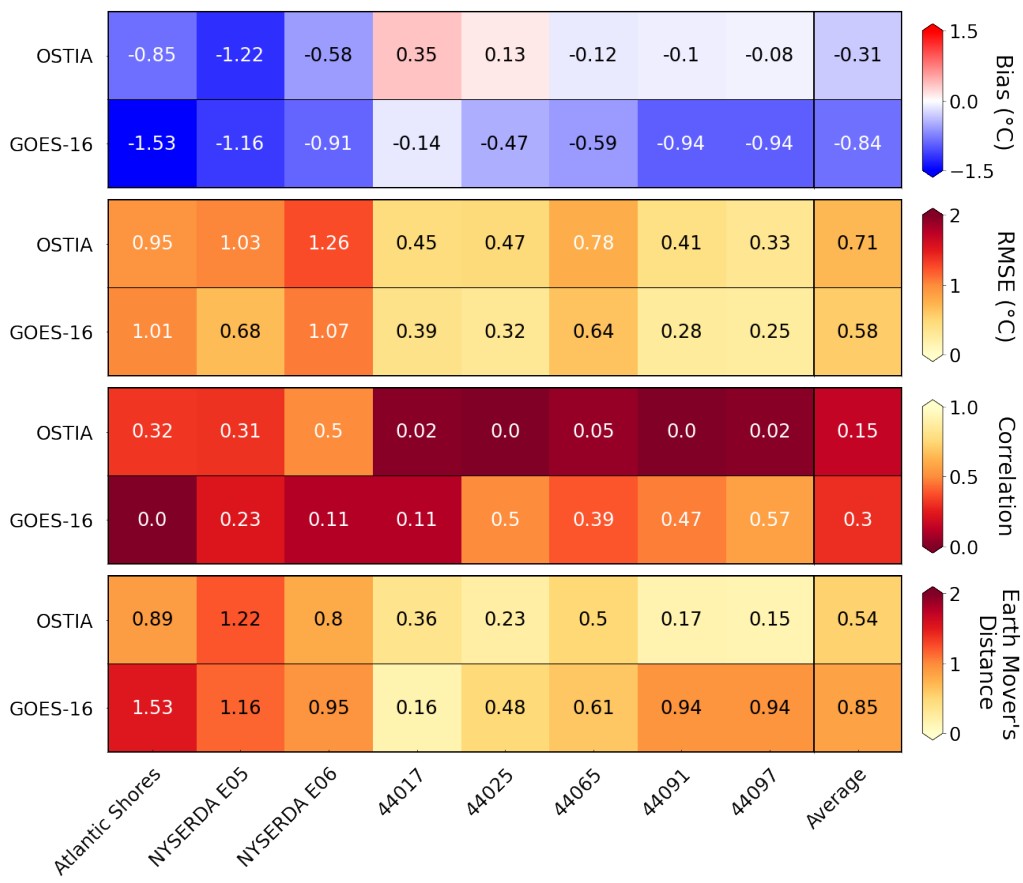

**Figure B1.** SST validation metrics (bias, RMSE, correlation, EMD) during the June 21-22 event, calculated across a subset of the buoy array. Omitted buoys are absent due to missing observational data at those locations.

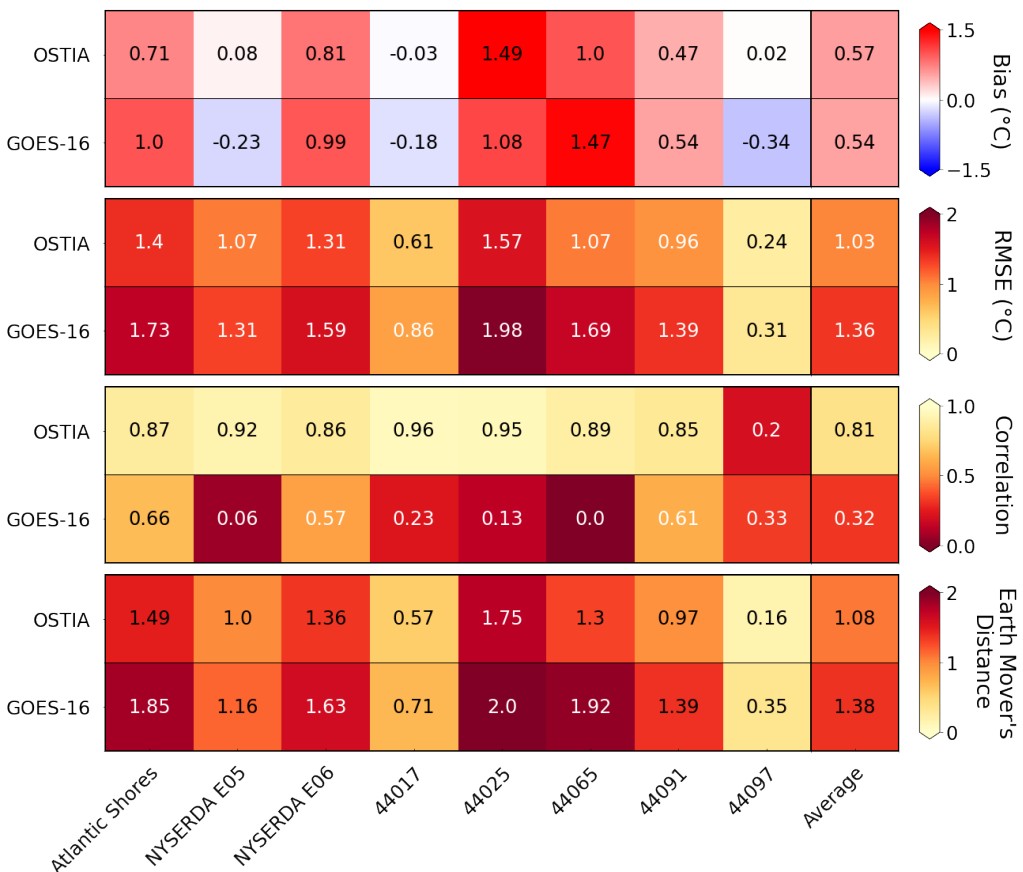

**Figure B2.** SST validation metrics (bias, RMSE, correlation, EMD) during the July 10-11 event, calculated across a subset of the buoy array. Omitted buoys are absent due to missing observational data at those locations.

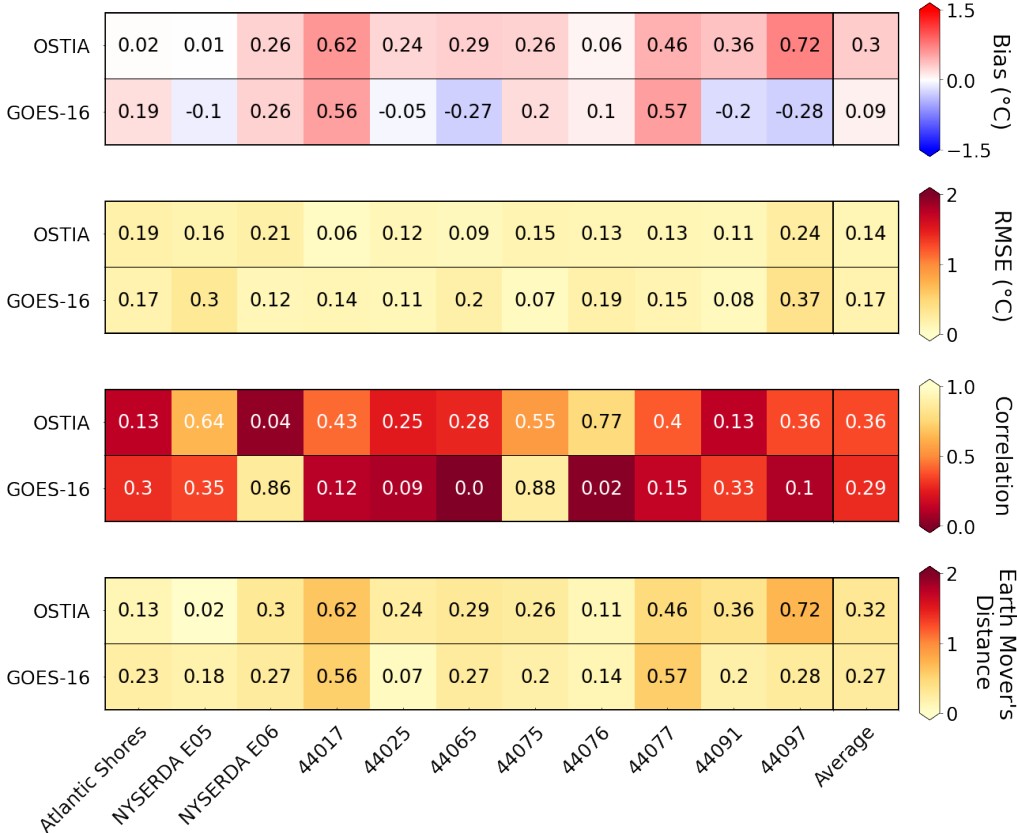

**Figure B3.** SST validation metrics (bias, RMSE, correlation, EMD) during the July 18 event, calculated across a subset of the buoy array. Omitted buoys are absent due to missing observational data at those locations.

## Appendix C:  Wind Speed Validation Heat Maps at 100 m and 120 m

Although this study considered a hub height of 140 m and therefore analyzed wind speeds primarily at that height, we include here validation metrics for each flagged event, at each lidar, for 100-m and 120-m hub heights. 140-m hub heights are expected to be close to the standard for offshore wind development (Lantz et al., 2019), but shorter turbines may also be constructed; additionally, having a better understanding of model performance across the rotor-swept area and not just at hub height can better inform wind power potential characterization.

Below, Fig. C1 presents 100-m model performance at the two NYSERDA lidars for the June 21-22 event, and Fig. C2 shows performance metrics for the same event and lidars at 120 m. At both heights, GOES-16 performs slightly better than OSTIA.

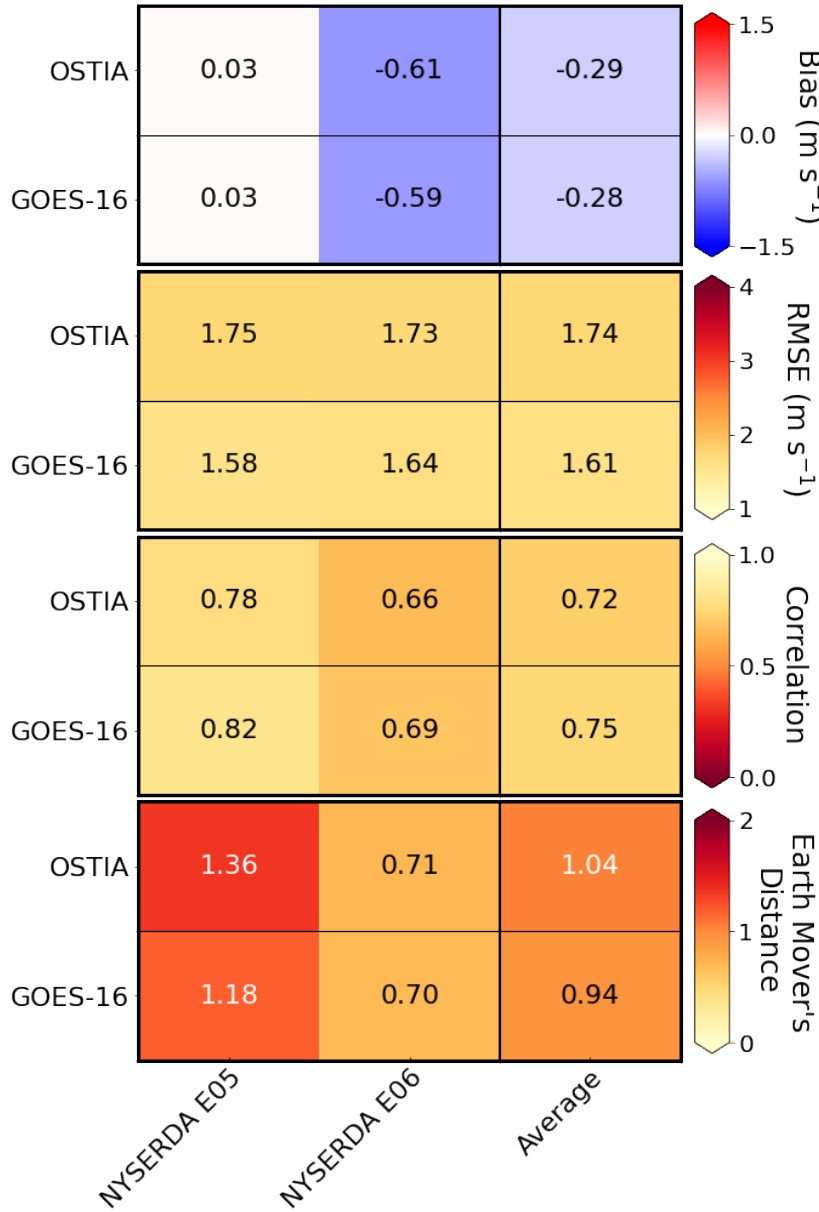

**Figure C1.** 100-m hub-height validation metrics at NYSERDA E05 and NYSERDA E06 during the June 21-22 event: bias, RMSE, correlation, and EMD.

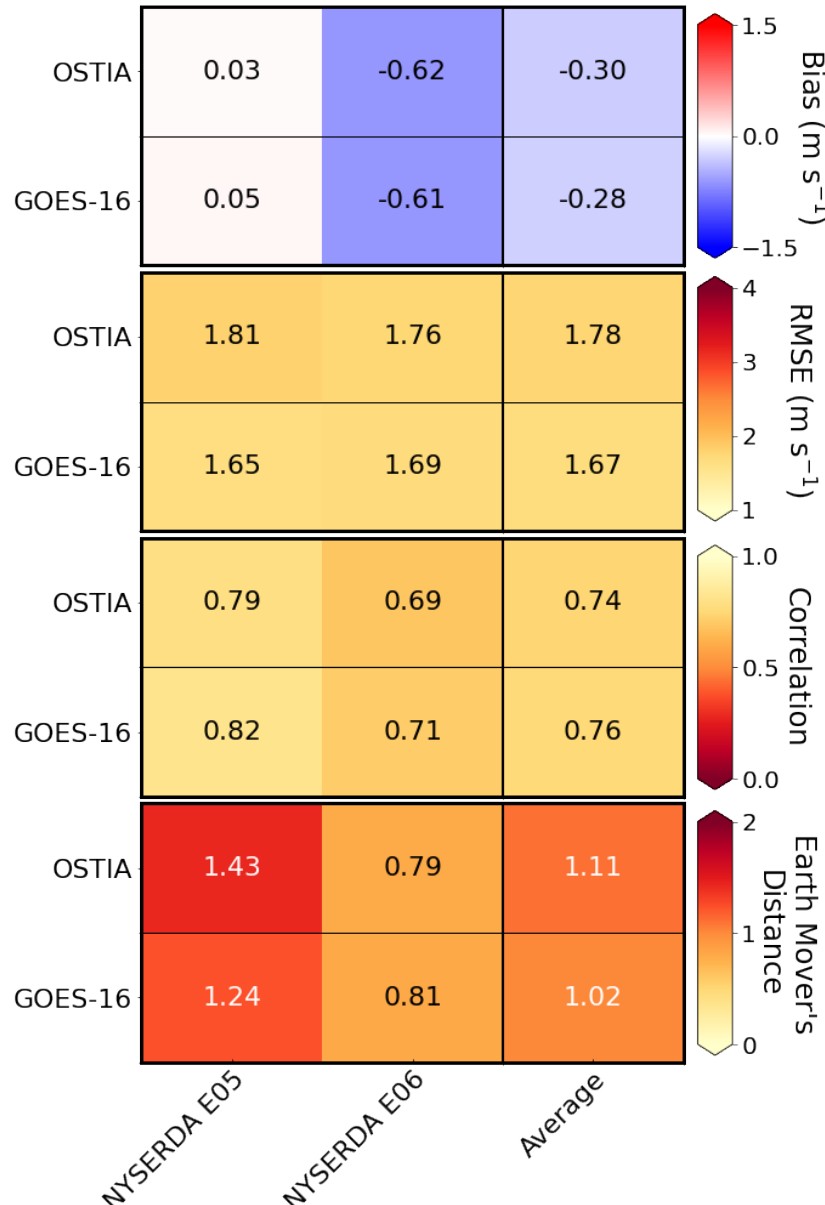

**Figure C2.** 120-m hub-height validation metrics at NYSERDA E05 and NYSERDA E06 during the June 21-22 event: bias, RMSE, correlation, and EMD.

Fig. C3 shows 100-m model performance at all three lidars for the July 10-11 event, and Fig. C4 presents performance metrics at all three lidars for the July 10-11 event at 120 m. While GOES-16 shows a higher average wind speed bias than OSTIA, the results from GOES-16 have a lower average RMSE and a higher average correlation than OSTIA. The wind speed

bias at all three lidars is notably strong, with a negative bias at Atlantic Shores (over 1 m s$^{-1}$ slower than observed) and a
405 positive bias at both NYSERDA lidars (well over 1 m s$^{-1}$ at a 120-m hub height at NYSERDA E05).

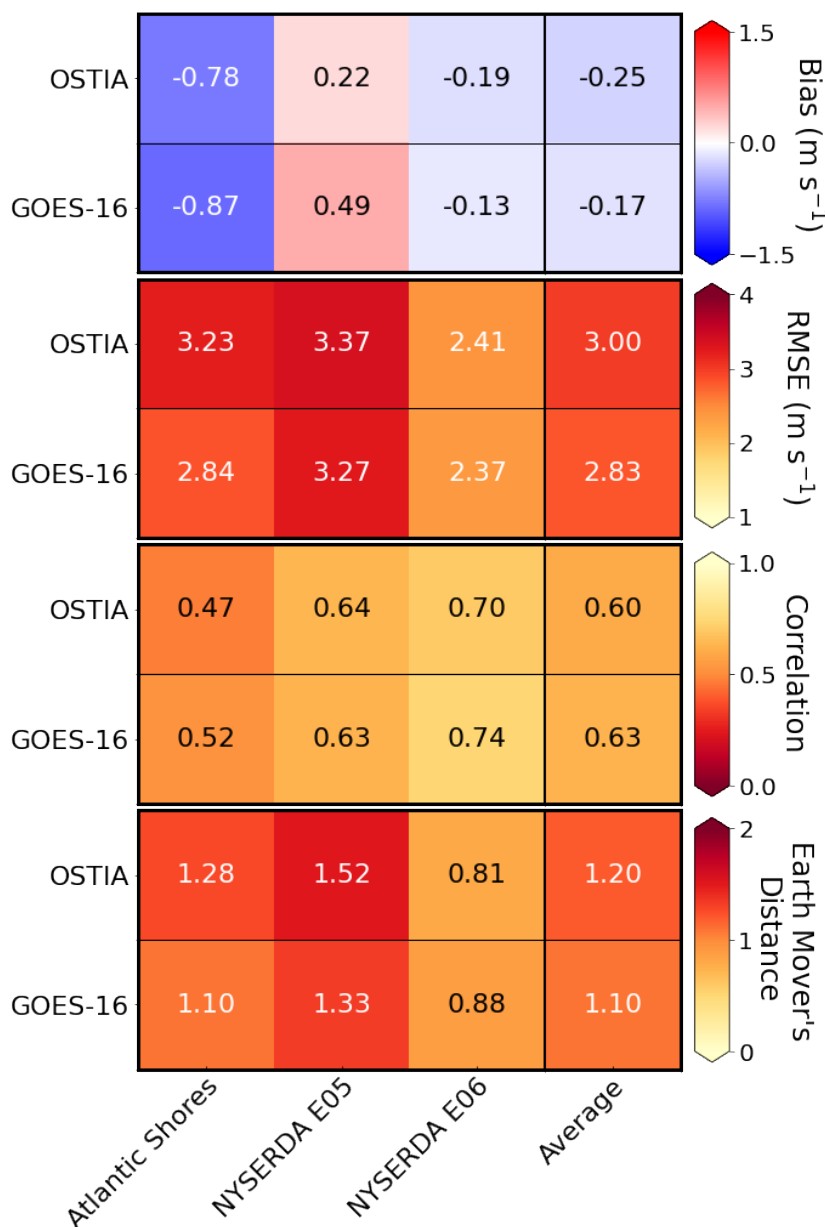

**Figure C3.** 100-m hub-height validation metrics at NYSERDA E05 and NYSERDA E06 during the July 10-11 event: bias, RMSE, correlation, and EMD.

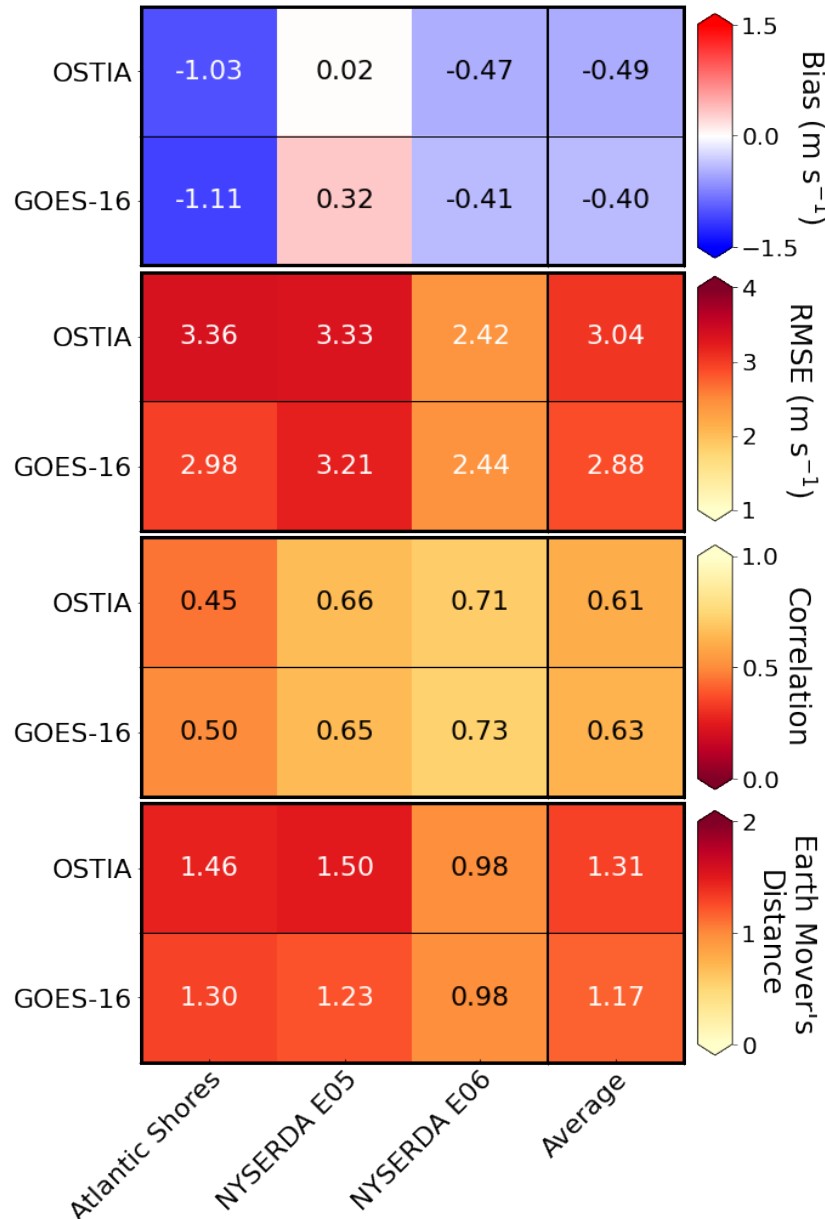

**Figure C4.** 120-m hub-height validation metrics at NYSERDA E05 and NYSERDA E06 during the July 10-11 event: bias, RMSE, correlation, and EMD.

Fig. C5 depicts 100-m model performance at Atlantic Shores and NYSERDA E06 for the July 18 event, and Fig. C6 shows performance metrics at Atlantic Shores and NYSERDA E06 for the July 18 event at 120 m. Although both simulations' hub-height wind speeds show comparable correlation with observations, OSTIA has an over 0.1 m s$^{-1}$ stronger wind speed bias than GOES-16, and GOES-16 validates with lower RMSE and EMD values.

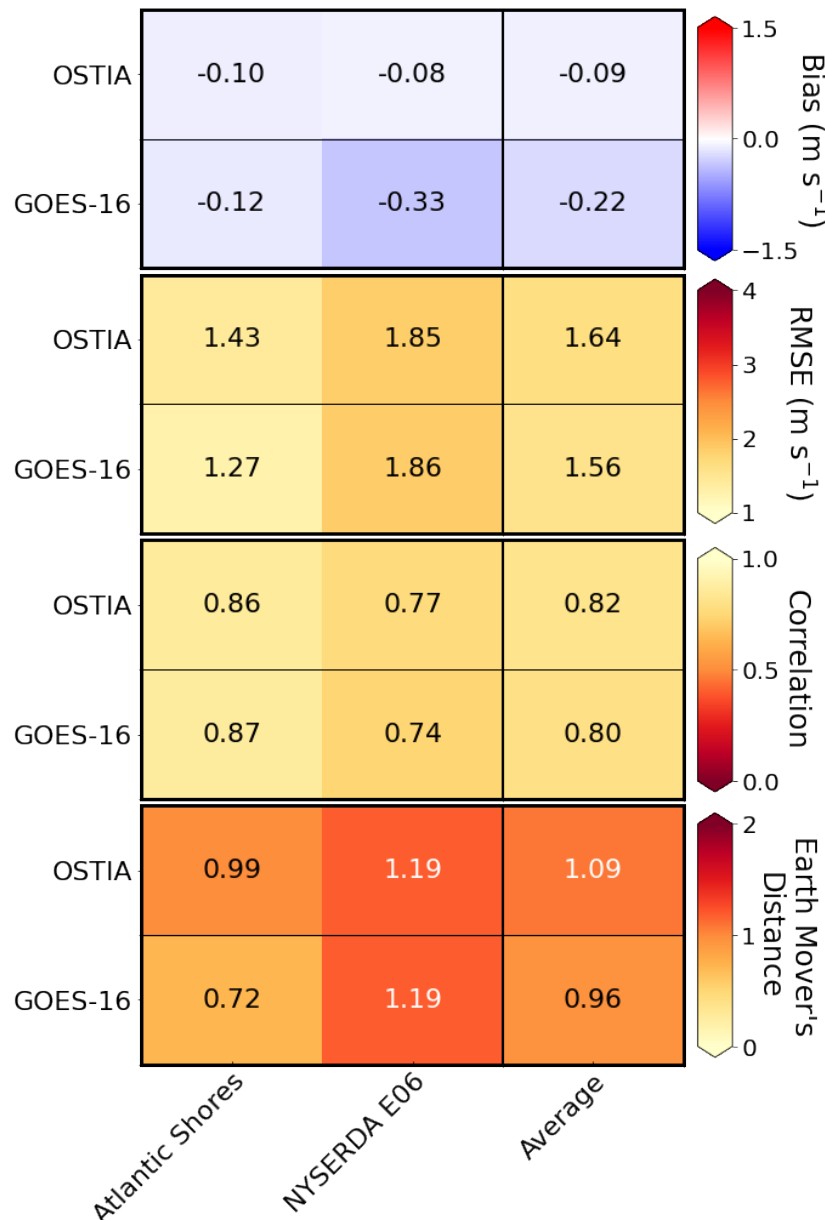

**Figure C5.** 100-m hub-height validation metrics at Atlantic Shores and NYSERDA E06 during the July 18 event: bias, RMSE, correlation, and EMD.

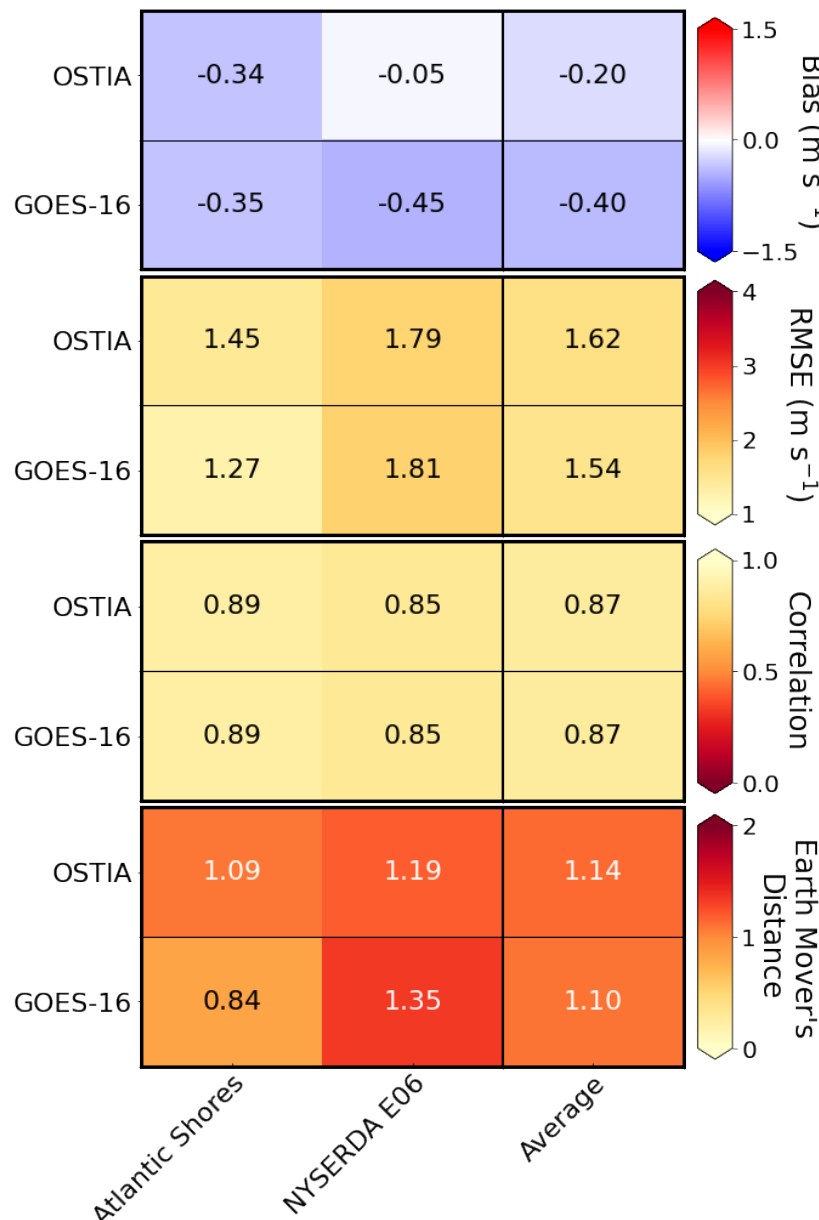

**Figure C6.** 120-m hub-height validation metrics at Atlantic Shores and NYSERDA E06 during the July 18 event: bias, RMSE, correlation, and EMD.

*Data availability.*

Buoy data outside of the Atlantic Shores and NYSERDA floating lidars were collected via the National Oceanic and Atmospheric Administration's NBDC (https://www.ndbc.noaa.gov). Satellite data were collected via the Physical Oceanography Distributed Active Archive Center. The WRF namelist may be obtained at https://doi.org/10.5281/zenodo.7275214.

*Author contributions.* SR downloaded and processed the GOES-16 data, ran the model for this SST input, and led the data analysis with
415 contributions from MO, GX, and CD. MO designed the model setup, ran the model with OSTIA data, and contributed to the data analysis.

*Competing interests.* The authors state that there is no conflict of interest.

*Acknowledgements.* We thank Patrick Hawbecker (National Center for Atmospheric Research) for providing code to overwrite ERA5 SSTs with those of each satellite product. We also thank Sarah C. Murphy (Department of Marine and Coastal Sciences, Rutgers University), Travis Miles (Department of Marine and Coastal Sciences, Rutgers University), and Joseph F. Brodie (Department of Atmospheric Research,
Rutgers University) for providing help and guidance with the DINEOF software.

This work was authored in part by the National Renewable Energy Laboratory, operated by Alliance for Sustainable Energy, LLC, for the U.S. Department of Energy (DOE) under Contract No. DE-AC36-08GO28308. Funding provided by the U.S. Department of Energy Office of Energy Efficiency and Renewable Energy Wind Energy Technologies Office and by the National Offshore Wind Research and Development Consortium under Agreement No. CRD-19-16351. The views expressed in the article do not necessarily represent the views of the DOE or
425 the U.S. Government. The U.S. Government retains and the publisher, by accepting the article for publication, acknowledges that the U.S. Government retains a nonexclusive, paid-up, irrevocable, worldwide license to publish or reproduce the published form of this work, or allow others to do so, for U.S. Government purposes. A portion of the research was performed using computational resources sponsored by the Department of Energy's Office of Energy Efficiency and Renewable Energy and located at the National Renewable Energy Laboratory.

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
