# Peer review of "Offshore wind energy forecasting sensitivity to sea surface temperature input in the Mid-Atlantic"

_Wind Energy Science, 2021_

## Author Response (AR1)

We thank the reviewers for taking the time to read our paper and provide thoughtful comments. We have made revisions as indicated below in the updated manuscript, and have addressed each comment below.

**Reviewer 1 Comments Response**

In this study, Redfern et al. compare the performance of WRF simulations forced by two SST products with different temporal, spatial and assimilation characteristics. They use lidar measurements as well as buoy measurements to compare the performance of the simulations. The evaluation is made through different error measures. They conclude that both SST products give comparable results on monthly time-scale, while during more challenging events GOES-16 outperforms OSTIA for simulating winds at 100 m height.

**General comments**

The study is interesting and certainly relevant for wind energy, since good wind forecast during challenging situations is critical. However, I think the analysis could be improved in terms of evaluating different heights. Most of the analysis focuses on hub height wind, which is defined as 100 m. However, the appendix shows, how much the error measures vary with height. Discussing other heights in more detail would create a more complete picture both with the prospect that hub heights of turbines are expected to increase as well as with respect to obtain a more complete picture for the entire rotor area.

Thank you for your comments. More details on metrics at different heights have been provided in the appendix. We have moved the hub height up to 140 m to be consistent with expected hub heights of future offshore installations, and have kept the focus on that level in the main manuscript, but have now also given analyses of performance at 100 m and 120 m in the Appendix.

I was confused about the third SST product, MUR. It is not very well introduced and only turnes up as a surprise for approximately 13 lines in the paper. It is neither mentioned in the abstract, discussion and conclusion. The role of MUR should be more clear and a proper introduction is necessary or it should be left out entirely. See my specific comment in that regard below.

Thank you for your comment. We have made changes to how MUR is introduced in the paper by addressing it in the abstract and clarifying sooner that its SST product is used

in our initial product comparison, but was removed from further analysis due to its poor validation against buoy measurements.

**Specific comments**

* Line 39: You could add the following paper to your discussion, which addresses the LLJ frequency in the area that you are investigating:
Aird, Jeanie A., Rebecca J. Barthelmie, Tristan J. Shepherd, and Sara C. Pryor. 2022. "Occurrence of Low-Level Jets over the Eastern U.S. Coastal Zone at Heights Relevant to Wind Energy" Energies 15, no. 2: 445. https://doi.org/10.3390/en15020445

Thank you. We have included the recommended reference in the manuscript on Lines 55-56.

* Line 81: Please add some more statistics about the availability of the buoys and lidars

We have added a list of owners of each buoy used in the study and a link in the Data Availability statement to the NDBC website where historical buoy data can be accessed.

* Line 90: In line 69 you write about two model simulations "We run two model simulations with identical setups, aside from the input SST data, off the Mid-Atlantic coast for June and 70 July of 2020.", while here you write about three simulations. "We compare how well three different SST datasets validate against buoy observations and subsequently select the two best- performing data sets to force our simulations (Table 3). Aside from these different SST product inputs, the rest of the model parameters in the simulations remain identical." In understand that you discard one SST product early on during you evaluation, but it is confusing as a reader to get this mixed information on how many simulations are performed. Also in table 3 there are only 2 products shown, which make it even more confusing. I see the first introduction to the third product, MUR, only during the analysis in section 3.1. I would suggest that you either always talk about three products or leave out the 3rd product. The least MUR should already be introduced in section 2.3.

Thank you for your suggestion. We have updated how we bring MUR into the conversation, by introducing it right away in the abstract and at the start of the Methods section.

* Line 154-156: Personally I don't like this style of summarizing the findings at the beginning of each section. It takes away the motivation to read the entire section. This is the same as in line 205-206.

Thank you for your comment. Respectfully, we are going to keep these summaries in the paper because the readers will need to read through the entire section to gather details about our findings. It is my understanding that many folks skim papers regardless, so we would prefer to make it easier for them to quickly locate our main points.

* Line 161: Linked to my point in line 90: Why introduce the 3rd data set with a "significant coarser" resolution here? It seems a bit unnecessary, especially in line 172 it is already disregarded from further analysis.

Thank you for this comment. We have adjusted the discussion around MUR to clarify its purpose in this study. Instead of throwing the results away, it seemed best to make them available for others in the case they might be useful for guiding future research.

* Line 171: According to Table 5 for Atlantic Shores, BIAS and EMD are rather comparable for MUR and the other products. You don't show the evaluation for the other stations for MUR. Based on Table 5, I find it a bit difficult to follow your assessment

Thank you for this comment. This table has been updated to reflect the validation metrics of SSTs at the buoys and all of the floating lidars for each product, during July only. The discussion has also been adjusted to explain that MUR's poor performance during July (one half of our focus period) is poor enough in relation to the other two datasets that we have removed it from further analysis.

* Line 179 - 181: I cannot follow your argument here: For instance, both for buoy 44017 and 44065 EMD, RMSE and Bias show worse performance for GEOS-16 than for OSTIA as well as compared to the average over all sites. Please clarify

Thank you for your comment. This has hopefully been clarified through a revision of this sentence in accordance with the new map in Figure 1 that now shows only the leasing sites (here, we now only consider the floating lidar buoys) on lines 88-90. In addition, we have included a new table showing the average performance metrics at these three lidars combined, over the two-month period in consideration.

* Line 187/188: It would be nice to see the boxplots to comprehend were your conclusions come from. If you don't feel there is a enough space to show all plots, it would be nice if you could add "(not shown)" to the text, so the reader does not continue searching for the plot related to that statement. This statement is also valid for line 171.

Thank you for your comment. These lines have been updated to include the phrase "(not shown)," in an effort to keep the number of figures included with the text at a reasonable level.

* Beginning of each section 3.3.1 - 3.3.3: To better understand the event, it would be good to have a description of the event at the beginning. This description could be what you have in line 225ff. This gives a good introduction to the event.

Thank you for this comment. To address this, the event sections have been revised to present synoptic conditions in the first paragraph.

* Line 215-220: As you motivate yourself 100 m is only one height. In figure 8 you could easily also add the matrices for e.g. 150 m height next to the matrices for 100 m.

Thank you for your comment. We have increased the hub-height for this revision by setting it at 140 m ASL, and cited the report motivating this decision (Lines 82-84). Because of this change, we do not believe that an additional matrix for 100-m wind speeds is needed in the main text; however, additional figures have been added to the appendix to show validation metrics at 100m and 120m.

* Line 239: An improvement for GOES-16 compared to OSTIA of 0.02 is indeed a very small. I would remove that statement.

Thank you for this comment. We have updated this paragraph per suggestions by both reviewers, and it now reads as follows:

"Over the course of June and July combined, looking across the entire buoy array, OSTIA overall outperforms GOES-16, as shown in Fig. 3. Both products have a relatively strong cold bias (between -0.1° C and -0.25°C) compared with observations at the three lidar locations. GOES-16 has a negative bias at five additional buoys, and OSTIA has a negative bias at one other buoy. Both SST products display warm biases at buoys 44017 (off the northeast coast of Long Island) and 44076 (the farthest offshore location considered, southeast of Cape Cod). Although GOES-16 follows the diurnal cycle rather than representing only the daily average SSTs, on average the correlation across all sites is comparable between the two SST products. RMSE is higher for GOES-16 at every buoy (with an average 0.64°C, compared with 0.52°C for OSTIA), and EMD for GOES-16 is higher than thot for OSTIA, at every location except the floating lidars (an average of 0.27, compared with 0.21 for OSTIA)."

\* Line 275: You show figures for wind direction in the above analysis, but you do not evaluate the performance of the different products in terms of wind direction or wind veer. Are they similar for those quantities?

Thank you for your commnet. We do not discuss wind direction and wind veer because they were beyond the scope of this study. Evaluating them has been listed at the end of the paper along with other future work avenues (Lines 345-346).

**Technical corrections**
Thank you for these comments.

\* Line 50: Please add citations for ERA5 and MERRA2

Citations for both of these datasets have been added in Table 2.

\* Line 62: Please add a citation for WRF

A citation for WRF has been included on line 96.

\* Figure 1: According to the WES guidelines: "A legend should clarify all symbols used and should appear in the figure itself, rather than verbal explanations in the captions (e.g. "dashed line" or "open green circles")". Please add a corresponding legend to the figure. Why are the leasing areas in different colours?

We have updated Figure 1 with a legend, and have removed the call areas (gray in the original figure) from the figure and instead have only plotted the leasing areas (white).

\* Table 2: Please consider to upload the wps and wrf namelists to a repository (e.g. zenodo) so the study becomes better reproducible

We have added into the data availability section that folks who are interested in obtaining specific namelists can email the author directly using the correspondence email for the paper.

\* Line 93: Please add a citation for OSTIA

We have updated the text to include a citation for OSTIA on line 112.

\* Line 98: Please add a citation for GOES-16

We have updated the text to include a citation for GOES-16 on line 116.

* Line 113: Please add a citation for DINEOF

A citation has been added at line 128.

* Figure 3: Are the buoys in a particular order? If not I would suggest to order them in an ascending order

We have reordered the buoys in this figure as suggested.

* Figure 4: It seems like you are not using the same bins for all of the three PDFs. This makes it difficult to compare the PDFs.

Thank you for pointing this out; we have updated the figure so that the bins are now all the same size.

* Figure 5: Why is 80 m highlighted although you mention in the text (line 191) that you consider 100 m as hub-height?

We have updated the horizontal line in this figure to match that in all the other figures with our new hub height of 140 m. Thank you for indicating this oversight.

* Figure 8: In contrast to figure 3 you did not reverse the colormap for correlation. So for figure 8 yellower colours are better for correlation, but worse for RMSE and EMD. I suggest to have better performance in the same colour for all matrices (this also goes for figure 11 and 14). Please show two significant digits for the bias for GOES-16, even if it is 0

We have updated the colormap with the noted reversal, and have also updated the figure to include two digits after each decimal point.

* Line 240: "although both present values very close to 0 m s-1" <- "are" is missing

Thank you for pointing this out. We have updated the text with results at a new hub height, so this sentence has been removed.

* Figure A1 - A3: hub height line is at 80 m instead of 100 m as stated in the description

We have updated the appendix figures to reflect the correct hub height (now updated to 140 m).

**Reviewer 2 Comments Response**

**General Comments:**

The paper of Redfern et al. addresses the topic of the role of SST representation in modelling wind energy forecast at offshore in the Mid-Atlantic. They compare the model and the observation wind speeds for the selected shorter-lived events (e.g., sea breezes and low-level jets). They examine the impact of SST on wind forecasting by using the OSTIA (daily) and GOES-16 SST (hourly) products in the model configuration.

The topic is of high interest in the context of studying the relationship between SST and offshore wind energy, and the paper is well-written. However, some clarifications and improvements are needed before the manuscript is publishable in the Wind Energy Source journal. Although the revision is somewhere between major and minor, I would like the authors to address all of my comments and suggestions that are listed below:

**Major Comments:**

Clarity of the abstract: I found the abstract is easy to read and understand the methodology for a reader. However, I would give a little more details, such as the horizontal resolution of the GOES-16 and OSTIA fields, the studying periods (June and July), and indicating that the wind speed deviations during flagged events are at 100 m hub-height. Concerning results, I would try to write some numerical results (e.g., better performance of xx% or so).

Thank you for your comments. We have updated the abstract with more detail, per the suggestions above, and hope that it now reads in more thoroughly.

The SST Performance: My feeling is that the SST analysis in this paper is not fully conducted on the event scale. Although the SST difference maps (e.g., Figure 9) are beneficial, they are not sufficient enough to track the daily fluctuations in SST at the event periods. I strongly suggest the authors show the time series of the SST products for all three lidars (not just the Atlantic Shores) and highlight the event dates on the time axis of these graphs. Additionally, the validation metrics of SST can be calculated for

each month, separately. I believe that these changes will help the reader to link the wind and SST event-scale changes.

Thank you for your comment. Unfortunately, zooming in on these short intervals, where the temporal resolution of the SSTs used in WRF are hourly (and maintain an hourly, stepwise pattern as well, in the output data), compared with the 10-minute resolution of the lidar, a lot of fidelity is lost (see below). With OSTIA in particular, since it is interpolated down to hourly intervals from a daily product, nuanced fluctuations in SSTs are not captured.

However, as suggested, the periods we selected are highlighted in light gray now on the new SST timeseries plots, and Table 5 has been expanded to include data from the other two lidars.

Then, I have other concerns. Discussion (and conclusion): what is the main take-home message? This point should be much more evident and clearer. I suggest that the authors can give more detail about the physical explanation of their outcomes (e.g., What might be the possible reason for these events correlate with the wind ramps?, Why do GOES-16 generally outperform OSTIA at different hub heights?). I would also compare the final results with similar studies and argue the uncertainties from several error sources (i.e., overall uncertainty in initial and boundary conditions, structural model uncertainty, etc.), and try to write some numerical results. In the conclusion, there is no need to explain the methodology in detail. A bullet point list may be helpful to summarize the main outcomes of the paper.

Thank you for your comment. We have addressed it by expanding on the Discussion section, condensing the conclusions, and including this paragraph at Line 350:

This study shows that SST inputs to WRF do affect forecasted winds and how well they validate against observations in  future leasing areas in the Mid-Atlantic. Although GOES-16 SSTs validate worse against buoy observations than OSTIA SSTs, on both monthly and event timescales, the wind characterization produced using GOES-16 inputs validates as well as, or in some cases better than, the OSTIA characterization, indicating that finer temporal and spatial SST resolution may contribute to improved wind forecasts in this region. To further explore this, a higher-quality post-processed GOES-16 data set assimilated with in situ observations, such as that created by Murphy et al. (2021), could be evaluated in future studies;  generating a comparable data set was beyond the scope of this study.

We have also included a new section in the Appendix with SST validation metrics for each of the events and discuss their relationship with 140-m wind speed metrics. Additionally, we have included in the manuscript the citations suggested by the reviewers.

**Minor Comments:**

1 Introduction:
        a)     P2L30: "The cold pool forms during the summer…" needs a reference.

We have added a reference on line 43-44.

    b)  P3L62: The authors can use the acronym 'NWP' instead of repeating the numerical weather prediction.

We have updated the manuscript to use the acronym instead of numerical weather prediction on this line.

2 Methods:
        a)     Why is August not included in this study? Wasn't there any short-lived event during August 2020? The authors can explain the reason in the manuscript.

August was not included in the study because at the time we began the study, we did not have observational data for that month. Unfortunately, at this time we are unable to run new simulations for August due to computational costs. We have added a brief note about this in the first paragraph of the methods section, on lines 80-82.

2.2 Model Setup:
    a)  P3L87: The resolution of the nested domain should be indicated in the text.

We have updated the manuscript on line 99 to indicate the nest resolution.

    b)     P4: 44008 station is not listed in Table 1.

We thank you for pointing this out and have added the buoy into Table 1.

    c)     P4: Please, indicate the horizontal resolution of the shown domain as well as the coordinates in Figure 1.

Figure 1 has been updated with these suggestions.

d)    P4: The units of coordinates should be indicated in Table 1.

Table 1 has been updated to show that coordinates are in degrees.

e)    P5: The physics schemes references are missing in Table 2 (e.g., RRTMG, Kain-Fritsch etc).

Table 2 has been updated to include this information and the associated references.

f)    Showing both domains (parent and nest) and their resolutions in one figure would be beneficial to explain the model setup.

We have updated Figure 1 to include an image showing the two WRF domains.

2.3. Sea Surface Temperature Data:
    a) P5L93: The OSTIA product needs a reference.
    b) P6L113: The DINEOF needs a reference.

Both of these references have been added, on lines 114 and 128.

2.4 Event Selection:
    a) Listing the event dates and simulation time in a table can help the reader to follow the methodology easily.

Per this suggestion, a new table (Table 7) has been added to the manuscript, which includes the event dates and lengths.

2.5 Validation Metrics:
    a) Why was particularly 100m hub height used in the evaluation? The authors can explain the reason in the manuscript.

We have updated our considered hub height to be 140 m instead of 100 m (lines 82-83).

Results:
3.1 Sea Surface Temperature Performance:

a)  The focus of this study is the GOES-16 and OSTIA SST products and their performance on the Mid-Atlantic coast. The MUR SST data is limited to the Atlantic Shores and not much successful in terms of catching the in-situ measurements of SST compared to the other two data sets during June and July of 2020. Is this data set truly needed for the SST analysis? Why?

We have primarily included MUR for posterity's sake, as we already had and were examining the data as part of a larger scale study, and including our analysis of it here may save others some time (and repeated work) in the future. Its poor SST performance is highlighted in the first step of our analysis, after which it is dropped from further evaluation.

We have updated the way that MUR is introduced and discussed in the text to clarify its purpose in our study.

b)      The correlation difference between the GOES-16 and OSTIA SST products in Figure 3 is not high. "Although GOES-16 follows the diurnal cycle rather than representing only the daily average SSTs, it still does not correlate with observations as well as the OSTIA" statement sounds like a strong judgment.

This whole paragraph has been updated on lines 182-189 in accordance with the suggestions presented here and in item c) below. It now reads as follows:

"Over the course of June and July combined, looking across the entire buoy array, OSTIA overall outperforms GOES-16, as shown in Fig. 3. Both products have a relatively strong cold bias (between -0.1° C and -0.25°C) compared with observations at the three lidar locations. GOES-16 has a negative bias at five additional buoys, and OSTIA has a negative bias at one other buoy. Both SST products display warm biases at buoys 44017 (off the northeast coast of Long Island) and 44076 (the farthest offshore location considered, southeast of Cape Cod). Although GOES-16 follows the diurnal cycle rather than representing only the daily average SSTs, on average the correlation across all sites is comparable between the two SST products. RMSE is higher for GOES-16 at every buoy (with an average 0.64°C, compared with 0.52°C for OSTIA), and EMD for GOES-16 is higher than thot for OSTIA, at every location except the floating lidars (an average of 0.27, compared with 0.21 for OSTIA)."

c)      The EMD performances of the SST products (Figure 3) also should be argued in the text.

Please see the response to item b) above.

3.2 Monthly Wind Speeds:
    a)  "Additionally, in both simulations, whole domain winds in July tend to be significantly faster than June winds." (P10L189) conflicts with the line "June average wind speeds are faster than those in July for both simulations." (P22L283) in the discussion section.

We thank the reviewer for catching this. The first sentence has been corrected at line 204, and it now reads as follows:

"Additionally, in both simulations, whole-domain winds in June tend to be significantly faster than July winds."

    b)  Why are the modeled hub-height wind speed bias and correlation for simulations in Figure 5 only shown for June 2020, not also for July 2020? The authors should state the reason in the manuscript.

This figure has been updated to include metrics over the whole two-month timespan, in what is now a 4x4 set of plots.

3.3. Event-Scale Wind Speeds:
    a)  P13L204: What are the criteria for the "little" in observational data?

This sentence has been clarified on line 219, and it now reads as follows:

"Lidars at which there are relatively large observational data gaps during these time periods (>20%) are not considered."

3.3.1. June 21 – 22, 2020 Event:
    a)  P15L226: Grammer mistake? ("affect")

Thank you for noting this grammatical error. We have fixed this in the text.

3.3.2. July 10 – 11, 2020 Event:
    a)  P16L235: "(Fig.10(a)))" one parenthesis is extra.

We have updated the text with the removal of this parenthesis.

\*\*  I suggest this reference concerning the sensitivity study of the WRF model (including the OSTIA SST) in offshore wind modeling in the Baltic  Sea: https://doi.org/10.1016/j.gsf.2021.101229.

We have included this reference in our manuscript on line 71.

---

## Author Response (AR2)

We thank the reviewers for their time and their feedback on the latest version of the manuscript and have provided responses to each of the comments below.

**Reviewer 1 Comments Response**

**General comments**

The authors have responded satisfactorily to most of my comments. Especially the third product, MUR25, is now better integrated into the overall flow of the manuscript. I think the manuscript can be accepted with minor revisions as stated below.

Thank you for your time, feedback, and suggestions. We have addressed each of them below.

**Minor Revisions**

- I think you might have misunderstood my comment "Line 81: Please add some more statistics about the availability of the buoys and lidars", which I would like to clarify: You state "There are periods of missing data for all buoys and lidars.", which is not very specific. Considering the statistics you show in e.g. Figure 3, it would be good to know, how the data availability is for each site within the study period June and July. For each of the events, you indeed highlight the availability for each lidar, but it would be good to see the overall data availability (e.g. 80% data availability for buoy 1 in June and July). This could be added to Table 1.

Table 1 has been updated with information about the percentage of data available at each lidar during the 2-month study period.

- Line 179 - 181: Did you apply linear interpolation? Please add.

We have noted that linear interpolation was used with the remotely sensed products (Line 179).

- Line 187 - 189: Why is the assessment for MUR25 only done for July 2020 and not for June, i.e., why do you need a separate Table 5 instead of adding MUR25 to Figure 3 to evaluate the performance for the entire study period?

When the manuscript was initially drafted, after evaluating SST performance we determined that MUR25 was a poor enough product to be dropped altogether from the wind forecasting analysis. Unfortunately, the WRF output was purged by our HPC team despite it being saved in what we had thought was a protected storage space. Because the SST validation metrics were calculated using the WRF output of SST, we can't do any further analysis on those data. What is

presented in the paper is what was originally done, and we kept it in because we hope it may be of use to others.

- Data availability: Having namelists directly accessible greatly reduces the barrier to reproduce the study and obtain details of your set-up. Thus, please add them as appendix or in an external repository, such as zenodo.

We have uploaded the namelist for the simulations to zenodo and listed the URL under the Data Availability section of the manuscript.

**Technical corrections:**

- Figure 1: is the border of (a) the area for domain 1? If not, could you add it to the figure? The rectangle marked with d02 in (a) does not completely agree with the extent of (b) according to the coastline, although according to the caption both represent the extent of d02. Please clarify. What does "The Atlantic Shores location is the site of both a buoy and a lidar." in the caption refer to? "Atlantic Shores" is the name of one site, but in total there are three sites with both lidar and buoy according to the map.

We have clarified in the figure caption that the more detailed map of the region (b) is a subset of the d02 domain showed in (a). We have also changed to caption to note that the NYSERDA sites also contain both a buoy and a lidar.

- Table 3: "Faily" -> replace with "Daily". Consider to add another row to the table highlighting that SSTs in MUR25 and OSTIA correspond to "foundation temperatures", while SSTs in GOES-16 correspond to "surface temperatures". This would help to see all differences between the different SST products at one glance.

Thank you for catching this typo; it has been fixed. We have also added as suggested another row to Table 3 listing the type of ocean temperature in each dataset.

- Line 114: I think the reference should be placed after 0.25°.

We have updated this sentence to include the reference after 0.25°.

- Line 232: Could you highlight "Rhode Island" on the map for any reader not familiar with US East coast geography?

This is a very good point. Figure 1 has been updated with state acronyms. Additionally, the reference in the text now states "…including a region planned for development just south of Rhode Island (Fig. 1, marked by "RI") …"

- Figure A1: Explain in the caption what G and O refer to. Although it is mentioned in the text above the figure, readers should be able to understand the figure from the figure and caption alone.

The captions for these figures have been updated with a statement about what G and O refer to.

- Figure 9 and similar figures (also in the appendix): Why is the range of the EMD between 0 and 3? There do not seem to be EMD values greater than 2, so the color scale could be re-scaled to show more variability. Please add units to bias and RMSE in all heatmap-figures.

All wind speed heatmaps have been updated with an updated EMD color range of 0 to 2, and units have been added to all heatmaps and relevant plots and tables.

**Reviewer 2 Comments Response**

**General Comments:**

The authors answered my comments well and made the corrections I pointed out. After adding the namelists as an appendix, I believe the manuscript is ready to be published in the WES journal.

We thank you for your time and feedback on our manuscript, and have uploaded the namelist to a zenodo archive so that others may be able to access it.